# Essential role of an ERV-derived Env38 protein in adaptive humoral immunity against an exogenous SVCV infection in a zebrafish model

Yun Hong[1], Chong-bin Hu[1], Jun Bai[1], Dong-dong Fan[1], Ai-fu Lin[1], Li-xin Xiang[1]\*, Jian-zhong Shao[1,2]\*

**1** College of Life Sciences, Key Laboratory for Cell and Gene Engineering of Zhejiang Province, Zhejiang University, Hangzhou, People's Republic of China, **2** Laboratory for Marine Biology and Biotechnology, Qingdao National Laboratory for Marine Science and Technology, Qingdao, People's Republic of China

\* xianglx@zju.edu.cn (LXX); shaojz@zju.edu.cn (JZS)

**Data Availability Statement:** The data that support the findings of this study are publicly available from [Dryad] with the identifier(s) [doi:10.5061/dryad. t4b8gtj5c].

## Abstract

Endogenous retroviruses (ERVs) are the relics of ancient retroviruses occupying a substantial fraction of vertebrate genomes. However, knowledge about the functional association of ERVs with cellular activities remains limited. Recently, we have identified approximately 3,315 ERVs from zebrafish at genome-wide level, among which 421 ERVs were actively expressed in response to the infection of Spring viraemia of carp virus (SVCV). These findings demonstrated the previously unrecognized activity of ERVs in zebrafish immunity, thereby making zebrafish an attractive model organism for deciphering the interplay among ERVs, exogenous invading viruses, and host immunity. In the present study, we investigated the functional role of an envelope protein (Env38) derived from an ERV-E5.1.38-DanRer element in zebrafish adaptive immunity against SVCV in view of its strong responsiveness to SVCV infection. This Env38 is a glycosylated membrane protein mainly distributed on MHC-II[+] antigen-presenting cells (APCs). By performing blockade and knockdown/knockout assays, we found that the deficiency of Env38 markedly impaired the activation of SVCV-induced CD4[+] T cells and thereby led to the inhibition of IgM[+]/IgZ[+] B cell proliferation, IgM/IgZ Ab production, and zebrafish defense against SVCV challenge. Mechanistically, Env38 activates CD4[+] T cells by promoting the formation of pMHC-TCR-CD4 complex via cross-linking MHC-II and CD4 molecules between APCs and CD4[+] T cells, wherein the surface subunit (SU) of Env38 associates with the second immunoglobin domain of CD4 (CD4-D2) and the first α1 domain of MHC-IIα (MHC-IIα1). Notably, the expression and functionality of Env38 was strongly induced by zebrafish IFNφ1, indicating that *env38* acts as an IFN-stimulating gene (ISG) regulated by IFN signaling. To the best of our knowledge, this study is the first to identify the involvement of an Env protein in host immune defense against an exogenous invading virus by promoting the initial activation of adaptive humoral immunity. It improved the current understanding of the cooperation between ERVs and host adaptive immunity.

**Funding:** This work was supported by the National Natural Science Foundation of China (32173003 to LXX; 31630083 to JZS) and the National Key Research and Development Program of China (2018YFD0900503 to JZS; 2018YFD0900505 to LXX). The funders had no role in study design, data collection and analysis, decision to publish, or preparation of the manuscript.

**Competing interests:** The authors have declared that no competing interests exist.

## Author summary

Endogenous retroviruses (ERVs) belong to an important subclass of transposon family that extensively exists in vertebrate and plant genomes. Understanding of the interplay between ERVs and host biological functions has long been a challenging frontier in life sciences. Fish is an indispensable integral part in this endeavor because it represents the most primitive host for retroviruses during vertebrate evolution. Here, by using a zebrafish model, we successfully identified the important role of an ERV-derived envelope protein (Env38) in adaptive humoral immunity against SVCV infection. We found that Env38 stimulates the initial activation of SVCV-induced CD4$^+$ T cells by cross-linking MHC-II and CD4 molecules during cell-cell interaction between MHC-II$^+$ APCs and CD4$^+$ T cells, thereby uncovering a novel stimulatory molecule in the central supramolecular activating complex (c-SMAC) of an immunological synapse structure. This finding indicated that Env38 has been co-opted for beneficial adaptive antiviral response by zebrafish immune cells. This study will greatly enrich current knowledge on the molecular mechanism underlying the activation of adaptive immunity and deepen understanding on the correlation of ERVs with exogenous invading viruses and host immune reactions.

## Introduction

Endogenous retroviruses (ERVs) are the relics of ancient retroviruses that occupy a substantial fraction (6%-14%) of all known vertebrate genomes [1,2]. These molecular remnants of once infectious retroviruses are previously considered as junk DNA or dark matter in host genomes because a vast majority of ERV elements are silenced by epigenetic modifications, and their interplay with host biological activities remains elusive [3–6]. However, studies are finding an increasing number of examples that ERVs have been co-opted for beneficial physiological functions by host cells [7]. A typical ERV contains *gag*, *pol*, and *env* genes encoding capsids, polymerases, and envelope proteins, which resembles those of exogenous retroviruses [8]. The envelope proteins (Envs) of ERVs have attracted considerably because of their extensive functional exaptation in a wide spectrum of biological processes, including growth, development, infection, immunity, and diseases [9]. Perhaps the best examples of ERV-derived Envs are Syncytin-1 and Syncytin-2 proteins, which play crucial roles in the implantation of embryos in the womb, placental morphogenesis, synaptic plasticity, and progression of multiple sclerosis (MS) in humans, mice, and other placental animals [10–12]. Additionally, preliminary investigations toward understanding the associations of ERV-derived Envs with host immune responses in human and mouse models have been conducted. Some Envs show immunomodulatory activity, such as Env derived from HERV-W induces proinflammatory cytokine expression in monocytes through the engagement of TLR4 receptor [13], and Env derived from HERV-K acts as a superantigen (SAg) to stimulate T cell-mediated B cell proliferation during EBV infection [14]. The mouse Env protein Fv4 restricts murine leukemia virus (MuLV) infection by competitively binding to cellular receptors [15]. The aberrant activation of HERV Envs is implicated in many pathophysiological processes, such as carcinogenesis and autoimmune diseases, including MS, systemic lupus erythematosus (SLE), rheumatoid arthritis (RA), insulin-dependent diabetes mellitus (IDDM), amyotrophic lateral sclerosis (ALS), and inflammatory neurologic disorders [16–20]. These observations uncover the role of ERVs as an unexpected ally in the regulation of host immune systems, suggesting that hosts accept the possible risks of detrimental immune reactions associated with the cooption and activation

of ERVs because the positive effects of ERVs may offset these risks [7]. Thus, understanding the interplay among ERV elements, particularly those of Envs, with host immune functions is becoming a new frontier in immunology and retrovirology.

Retroviruses exclusively infect almost all vertebrate species, among which fish is believed to be the most primitive vertebrate host for retroviruses [21]. In fact, one of the oldest intact ERV *env* genes with functional exaptation is reported to be the *percomORF* gene of ray-finned fish, which has undergone a long-term purifying selection for more than 100 million years [22]. Developments in high-throughput sequencing technology has facilitated the discovery of abundant ERVs with coding potential in many fish species [23]. Thus, fish is integral to the understanding of the molecular and functional evolutionary history of the ERV superfamily. In line with this notion, we recently performed a genome-wide characterization of ERVs in zebrafish (*Danio rerio*) from multiple perspectives, including composition, genomic organization, classification, phylogeny, and expression profiles in embryos and adult tissues under physiological and virus infection conditions [24]. Approximately 3,315 ERV elements were identified from all the 25 chromosomes of zebrafish, which were classified into Gypsy, Copia, Bel, and Class I-III groups. Among them, 665 ERVs were actively expressed in embryos and tissues under physiological conditions, and the majority of these ERVs exhibited stage and tissue specificity. Importantly, 421 ERVs were remarkably induced in immune-related tissues, including the head kidney, spleen, and gut in response to Spring viraemia of carp virus (SVCV) infection. Most of these ERVs were rarely activated in tissues at steady state without SVCV stimulation, indicating their strong responsiveness to virus infection. A total of 71 *env* genes were detected. These Env elements with TLV coat (HR1–HR2) domains were phylogenetically associated with Class I ERVs and categorized into four subtypes (*Dr*Env1–4). Among which, *Dr*Env1 and *Dr*Env4 show high degrees of homogeneity and are likely derived from the same ancestor, whereas *Dr*Env2 and *Dr*Env3 show heterogeneity and seem to be correlated with exogenous retroviruses (XRVs) and mammalian ERVs. These findings demonstrated the previously unrecognized abundance and immunological activity of ERVs in zebrafish, thereby making it an attractive model organism for deciphering complex interplay among ERVs, exogenous invading viruses, and host immunity. In the present study, we investigated the regulatory role of an *Dr*Env3-typified envelope (Env) protein in the adaptive antiviral immunity against SVCV in zebrafish in view of its strong response to SVCV infection [24]. The Env protein was derived from an ERV-E5.1.38-DanRer element located on zebrafish chromosome 5 and named ERV-E5.1.38-DanRer-Env (Env38 in brief). The results showed that Env38 is essential to the full activation of CD4$^+$ T cell-mediated adaptive humoral immunity against SVCV. To the best of our knowledge, this study is the first to identify the role of an ERV-derived envelope protein in adaptive immune defense against an exogenous invading virus, which uncovered an unexpected functional mechanism of ERV-encoded Env proteins in antiviral immunity by promoting the initial activation of adaptive humoral immune response.

## Results

### Molecular characterization of ERV-E5.1.38-DanRer-Env (Env38)

By RetroTector prediction, a typical ERV that possesses complete structure with *gag*, *pol*, and *env* genes and LTRs at both ends was retrieved from a *si:zfos-375h5.1* locus on zebrafish chromosome 5 (Fig 1A). In this prediction, the sequence of chromosome 5 was segmented into 9 Mb fragments with 2 Kb overlapping regions between fragments, and the predicted ERV was named ERV-E5.1.38-DanRer according to the nomenclature rules described previously [24,25]. In ERV-E5.1.38-DanRer, "E" represented epsilon-related lineage, "5" represented chromosome 5, "1" meant the first 9 Mb fragment of chromosome 5, and "38" referred to the

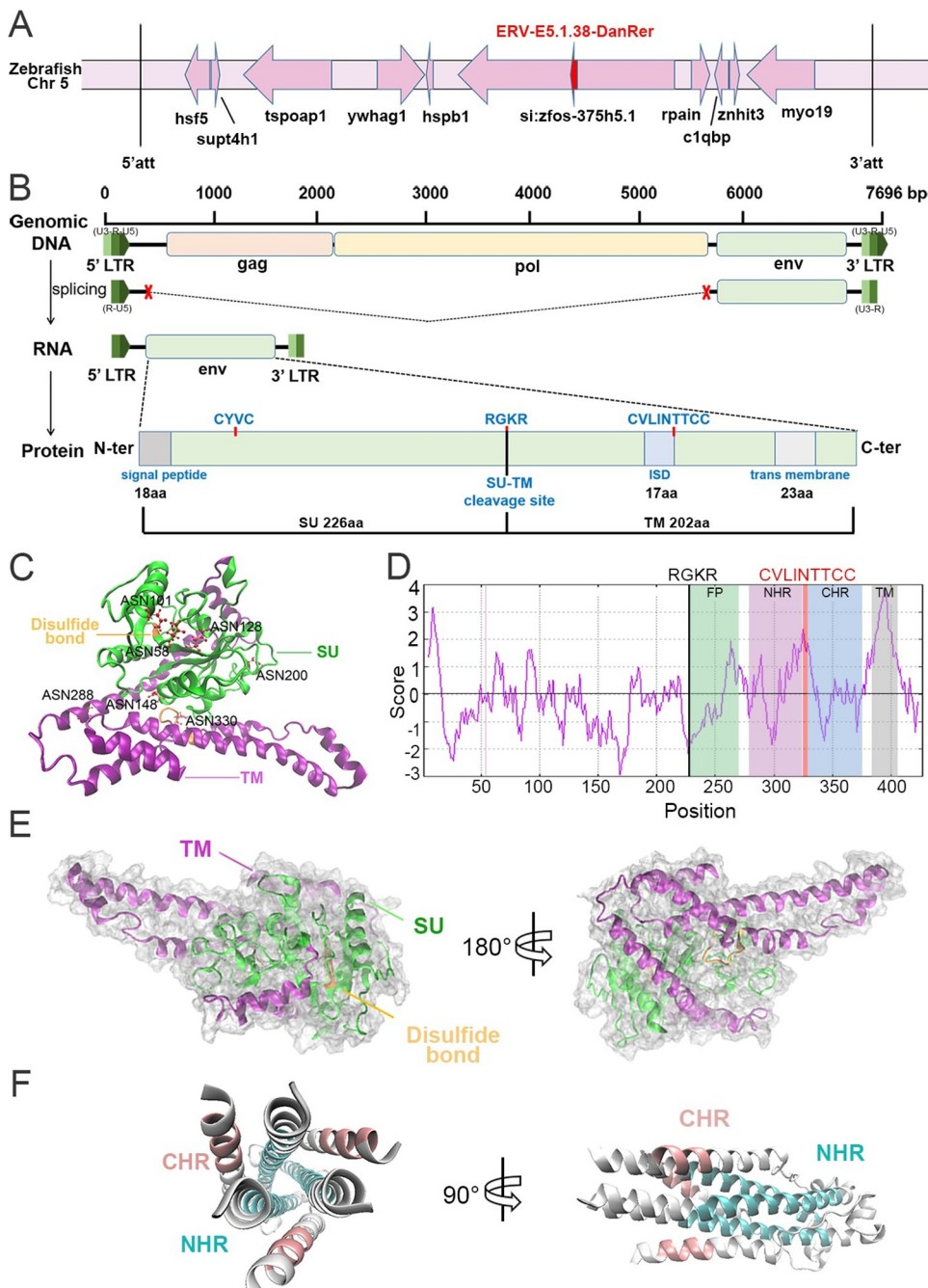

**Fig 1. Bioinformatic characterization of ERV-E5.1.38-DanRer element and Env38 molecule.** (A) Genomic localization of ERV-E5.1.38-DanRer element in zebrafish. The locus of ERV-E5.1.38-DanRer element on zebrafish chromosome 5 is indicated in red arrow. The ERV-E5.1.38-DanRer is distributed inside *si:zfos-375h5.1*. The upstream *hspb1*, *ywhag1*, *tspoap1*, *supt4h1*, *hsf5* and downstream *rpain*, *c1qbp*, *znhit3*, *myo19* genes of ERV-E5.1.38-DanRer are presented, and att means attachment site. (B) Schematic diagram showing the transcription and splicing of *env38* gene, as well as the structural characterization of Env38 protein, in which a signal peptide (1–18 amino acid), a furin cleavage site (R/K-X-R/K-R, 227–230 amino acid), an immunosuppressive domain (ISD, 309–325 amino acid), a transmembrane (TM) domain (383–405 amino acid) and conserved C-X-X-C and C-X5/6/7-C motifs are presented. The LTR consists of U3, R, and U5. The transcription of *env38* begins at 5'-LTR-R and ends at 3'-LTR-R. (C) The five predicted N-glycosylation sites in SU subunit and two in TM subunit are marked with ASN residues. SU: surface subunit; TM: transmembrane subunit. (D) Hydrophobicity analysis of Env38 protein showed that the fusion peptide next to the furin-cleavage site was partial hydrophobic. FP: fusion peptide; NHR: N-terminal heptad repeats; CHR: C-terminal heptad repeats; TM: transmembrane domain. (E) Tertiary structure of Env38 monomer modeled by

I-TASSER. Prominent central α-helical coiled-coil (CC) structures were found in the TM subunit of Env38 monomer protein. The top five threading templates were 6vkm, 6rx3A, 6vkm, 6fgzA, and 5yfpH. TM: transmembrane subunit; SU: surface subunit. (F) Tertiary structure of the NHR-CHR trimeric domain of Env38 TM subunit predicted by SWISS-MODEL program with crystal structures of human Syncytin 1 in post-fusion conformation (SMTL ID: 6rx1.1) as a model. A homotrimer with a fusion core structure was formed by three NHR-CHR domains in which three helical NHR peptides formed a central core and three helical CHR peptides packed into the grooves on the surface of the central core.

38th ERV located in the first 9 Mb fragment [24]. Thus, ERV-E5.1.38-DanRer represented the 38th ERV distributed in the first segmented 9 Mb fragment on chromosome 5. Accordingly, the Env element of ERV-E5.1.38-DanRer was named ERV-E5.1.38-DanRer-Env, and this name was abbreviated as Env38 for simplification. The *env38* was an intronless gene (GenBank accession number: ON420216), which consisted of 7,696 bp and distributed downstream *gag* and *pol* genes. The full-length cDNA of *env38* consisted of 1,843 bp, which contained a 196 bp 5′UTR (including R, U5 and noncoding regions), a 1,287 bp open reading frame (ORF) that encoded 428 amino acids, and a 360 bp 3′UTR (including U3, R and noncoding regions) (S1A Fig). *env38* was transcribed in the same manner as exogenous retroviruses in which the transcription started from the 5′ end of the R region in 5′-LTR and terminated at the 3′ end of the R region in 3′-LTR, and the majority of sequences between 5′-LTR and *env* gene was spliced, including the ORFs of the *gag* and *pol* genes. Consequently, the mature transcript of *env38* retained the R and U5 regions at the 5′ end and U3 and R regions at the 3′ end, in which only the ORF of *env* and two noncoding regions at both ends of *env* were kept (Fig 1B).

The Env38 protein consisted of 428 amino acids with a putative molecular weight of 48 kDa. By using the CDD of NCBI carried by retroviral Envs [24], Env38 protein was predicted to be a type I membrane protein with an N-terminal signal peptide (1–18 amino acid), a furin cleavage site (R/K-X-R/K-R, 227–230 amino acid), an N-terminal heptad repeat (NHR, 286–314 amino acid), an immunosuppressive domain (ISD, 309–325 amino acid), a C-terminal heptad repeat (CHR, 351–358 amino acid), a transmembrane (TM) domain (383–405 amino acid), and a C-terminal cytoplasmic tail (CT, 406–427 amino acid). The Env38 protein potentially contained a surface subunit (SU, 226 amino acids) and a transmembrane subunit (TM, 202 amino acids) given the existence of a furin cleavage site in the precursor Env38 protein. The C-X-X-C and C-X5/6/7-C motifs involved in disulfide bond formation in SU–TM interaction were predicted in SU and TM domains, respectively (Figs 1B and S1A). The SU and TM subunits contained five and two potential N-glycosylation sites, respectively (Figs 1C and S1A). A partial hydrophobic fusion peptide was predicted next to the furin cleavage site, suggesting the inefficient fusion function of Env38 protein (Figs 1D and S1B). Additionally, I-TASSER program was used to model the domain architecture and tertiary structure of Env38 protein. Prominent central α-helical coiled-coil (CC) structures were found in the TM subunit of an Env38 monomer protein (Fig 1E). These α-helical CC structures are typically observed in the fusion subunits of viral class I fusion protein family members [26], suggesting that Env38 belongs to a class I fusion protein. Given the extreme dissimilarity in SU sequence between Env38 and other viral envelope proteins, SWISS-MODEL program was used to predict the tertiary structure of the conserved NHR–CHR domain in Env38 TM with the crystal structure of human Syncytin-1 in post-fusion conformation (SMTL ID: 6rx1.1) as a model [27]. The result showed that three NHR–CHR domains formed a homotrimer with a fusion core structure in which three helical NHR peptides formed a central core and three helical CHR peptides packed into the grooves on the surface of the central core (Fig 1F). The homotrimer structure with six helix bundles is the conservative feature of class I fusion proteins [26].

To provide experimental verification, intact Env38 proteins with a His-tag were expressed in *E. coli* and Sf9 cells, respectively. The recombinant Env38 proteins were prepared by affinity purification and detected by 12% SDS-PAGE. The Western blot analysis showed an expected molecular weight of approximately 48 kDa in *E. coli* and an unexpected molecular weight of approximately 60 kDa in Sf9 cells (Fig 2A). Increments in the molecular weights of Env38 protein in eukaryotic cells suggested the potential glycosylation of Env38 protein like that in viral envelope proteins. This hypothesis was evaluated by PNGase F digestion on the mature Env38 protein expressed in Sf9 cells and tunicamycin inhibition on the glycosylation process of Env38 protein in HEK293T cells transfected with pcDNA3.1-Flag-Env38-LTR [28]. As expected, a shed protein with a molecular weight of 48 kDa was detected by SDS-PAGE and Western blot after PNGase F and tunicamycin treatments. In these cases, other shift bands might reveal unequal degrees of digestion and deglycosylation (Fig 2B). As a support, Sf9 cells-expressed Env38 protein exhibited a strong positive stain in SDS-PAGE gel after treatment with Schiff reagent, which is widely used in detecting polysaccharides and glycoproteins [29,30]. By contrast, *E. coli* cell-expressed Env38 protein showed no stain (Fig 2C). The furin cleavage potential of Env38 protein was examined in HEK293T cells by coexpression of Env38 and zebrafish FurinA or FurinB enzyme (S2 Fig). The results showed that the Env38 protein can be cleaved by zebrafish furin enzymes into SU and TM subunits with an expected molecular weight of 35 kDa and 25 kDa, respectively (Fig 2D and 2E). However, no cleavage of the Env38 protein was detected in the spleen, head kidney, and leukocytes of zebrafish stimulated with SVCV ($10^5$ TCID$_{50}$) (Fig 2F). In this case, the expression of *furina* and *furinb* was undetectable in Env38$^+$ cells sorted from leukocytes of spleen, head kidney and peripheral blood (Fig 2G). These findings implied that Env38 functioned in non-cleavage and cleavage manners, depending on the differential expression of FurinA or FurinB in cells correspondingly.

## Preparation of polyclonal antibody, siRNA-encoding LV and knockout zebrafish

Polyclonal antibody (Ab) against Env38 (anti-Env38) was prepared from immunized mouse serum into IgG isotype through protein-A affinity purification as described previously [31,32]. The reactivity and specificity of Ab in targeting the Env38 proteins were determined by ELISA and Western blot analysis. The target protein samples were derived from *E. coli* and HEK293T cells ectopically expressed with recombinant Env38 protein and zebrafish spleen, head kidney, leukocytes naturally expressed with Env38 protein. The results showed that the anti-Env38 Ab exhibited high reactivity and specificity to recombinant and natural Env38 proteins with an average titer above 1:10,000 according to ELISA analysis. Anti-Env38 Ab clearly combined with target Env38 proteins with molecular weights of approximately 48 kDa and 60 kDa, respectively (Fig 2F). For generating siRNA-encoding LVs, three candidate siRNAs (siRNA1, siRNA2 and siRNA3) targeting different regions of *env38* mRNA were designed, and siRNA1 had the highest efficacy in inducing *env38* mRNA degradation (S3A Fig). This siRNA-1 was then used in constructing an siRNA-encoding LV (Env38siRNA-LV). The titer of Env38siRNA-LV was $3 \times 10^{10}$ TU/ml when assessed in HEK293T cells by EGFP-based FCM analysis (S3B and S3C Fig). The interference activity of Env38siRNA-LV was examined by knockdown evaluation *in vivo*. Zebrafish were i.p. administered with Env38siRNA-LV ($1 \times 10^6$ TU/fish) or scrambled siRNA-encoding LV (contsiRNA-LV) three times at 24 h intervals after SVCV ($10^5$ TCID$_{50}$) stimulation. Seven days after SVCV stimulation, the expression levels of Env38 mRNA and protein in leukocytes from peripheral blood, head kidney and spleen tissues and Env38 protein in spleen tissues were significantly downregulated, as determined by RT-qPCR, FCM analysis and Western blot (S3D–S3G Fig), among which the percentage of Env38$^+$ cells

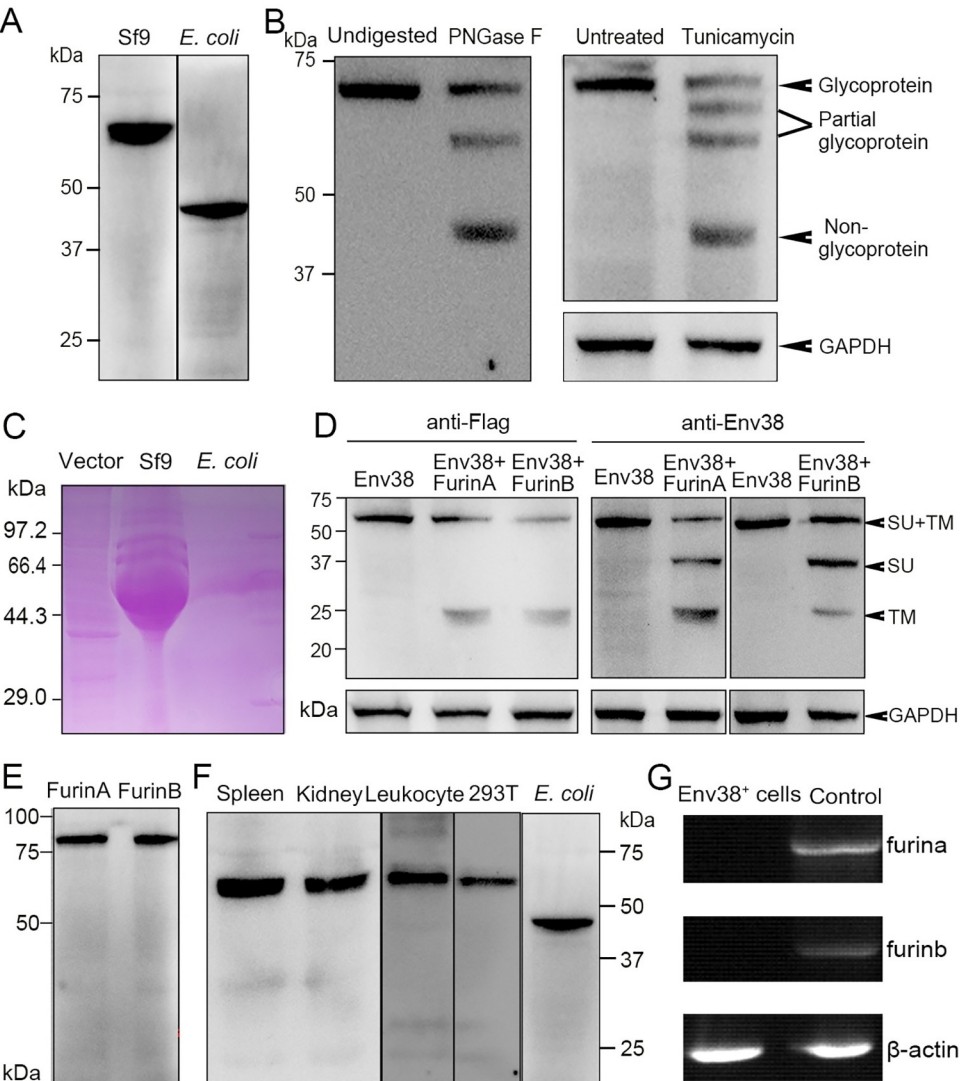

**Fig 2. Examination on the expression, glycosylation and proteolysis of Env38 protein and activity of anti-Env38 Ab.** (A) Western blot analysis for the expression of Env38 protein with mouse anti-His Ab (1:5,000). The Env38 proteins expressed in eukaryotic Sf9 cells and prokaryotic *E. coli* cells were determined to be approximately 60 kDa and 48 kDa, respectively. (B) Western blot analysis for N-glycosylation of Env38 protein with treatment of PNGase F and tunicamycin, in which the Env38 protein derived from Sf9 cells was treated with PNGase F (500 U/ml), and the Env38-expressing HEK293T cells transfected with pcDNA3.1-Flag-Env38-LTR (0.6 μg/ml) was treated with tunicamycin (5 μg/ml). The Env38 protein was separated by 12% SDS-PAGE. Env38 protein derived from Sf9 cells without digestion and HEK293T cells without treatment were served as controls. GAPDH was used as a loading control. (C) Detection of the glycosylation of Env38 protein by PAS reaction. The Env38 proteins were expressed in Sf9 cells or *E. coli* cells, and separated by 12% SDS-PAGE followed by staining with Schiff reagent. Env38 protein derived from Sf9 cells exhibited a strong positive staining, while Env38 protein derived from *E. coli* cells showed no staining. (D) Western blot analysis for the detection of enzymatic cleavage of Env38 protein by Furins. Coexpression of Env38 protein (with a Flag-tag at C-terminus) and FurinA or FurinB in HEK293T cells resulted in cleavage of Env38 with two protein bands corresponding to the intact Env38 protein (60 kDa) and TM subunit (25 kDa) when detected with anti-Flag Ab, and three protein bands corresponding to the intact Env38 protein (60 kDa), SU subunit (35 kDa) and TM subunit (25 kDa) when detected with mouse anti-Env38 Ab. GAPDH was used as a loading control. (E) Western blot analysis for the expression of FurinA and FurinB proteins in HEK293T cells. (F) Western blot analysis for the reactivity and specificity of mouse anti-Env38 polyclonal Ab. The anti-Env38 Ab clearly combined with the target natural Env38 proteins derived from spleen, head kidney and leukocytes, as well as recombinant Env38 proteins from HEK293T and *E. coli* cells with molecular weights of 60 kDa and 48 kDa, respectively. (G) Detection of the transcriptional expression of *furina* and *furinb* genes in Env38+ cells sorted from leukocytes from spleen, head kidney and peripheral blood of zebrafish stimulated with SVCV ($10^5$ TCID$_{50}$) using RT-PCR. Total leukocytes were used as a control.

decreased from 28.90% ± 1.32% to 5.08% ± 0.20% in leukocytes of fish received contsiRNA-LV and Env38siRNA-LV (S3E Fig). For the generation of Env38 knockout zebrafish (Env38$^{-/-}$ zebrafish), the targeted "C" of the 367th base was replaced with the sequence "TTCAAGGCT" (S4A Fig), and the Env38$^{-/-}$ zebrafish homozygote was acquired after two generations. Western blot was performed to show the absence of Env38 protein in Env38$^{-/-}$ zebrafish (S4B Fig).

## Subcellular localization and tissue/cellular distribution of Env38

For subcellular localization analysis, the EGFP-fusion Env38-encoding plasmid (pEGFP-N1-Env38) was transfected into HEK293T cells. The Env38-EGFP signal was mainly distributed in the plasma membrane and cytoplasm of HEK293T cells (Fig 3A). Moreover, when transfected cells were stained with fluorescent probes for endoplasmic reticulum (ER-Tracker Red) and Golgi apparatus (Golgi-Tracker Red), the Env38-EGFP signal merged with the ER-Tracker Red or Golgi-Tracker Red signal (Fig 3B and 3C). These results indicated that the cytoplasmic Env38 protein was localized on ER and Golgi apparatus. Similar results were observed by detecting non-EGFP tagged Env38 with Env38-specific antibody (S5 Fig). Because EPC cells were highly sensitive to SVCV infection, they were used for detecting the subcellular localization of Env38 in response to SVCV stimulation. The results showed that the Env38 proteins were sporadically detected on plasma membrane of EPC cells in the absence of SVCV (Fig 3D). However, considerable Env38 proteins translocated onto the plasma membrane of rounded EPC cells upon stimulation with SVCV (Fig 3E). This finding implied that Env38 might act as a trafficking membrane protein that translocates from ER and Golgi apparatus onto plasma membrane after SVCV stimulation. In addition, MHC-II$^+$Env38$^+$ and IgM$^+$Env38$^+$ cells were observed in the leukocytes of zebrafish with SVCV stimulation (Figs 4, S6A and S6B). However, Env38 was hardly detected on/in unpermeabilized and permeabilized control leukocytes without SVCV stimulation (S6C and S6D Fig). These observations indicated the induced expression and translocation of Env38 upon SVCV infection, suggesting the functional role of Env38 in SVCV-induced antiviral immunity of zebrafish.

Next, the dynamic tissue/cellular distribution of Env38 in response to SVCV infection was examined to provide initial insights into the role of Env38 in antiviral immunity. RT-qPCR results showed that the expression of *env38* was detected in the liver, brain, heart, intestine, skin, and gill at various degrees under normal conditions, suggesting the extensively involvement of Env38 in multiple physiological activities at steady state of zebrafish without viral stimulation (Fig 5A). However, after stimulation with SVCV, the transcriptional expression of *env38* was significantly upregulated in most immune-related tissues, including the spleen, head kidney, intestine, skin, and gill, and leukocytes from spleen, head kidney and peripheral blood (Figs 5A and S7). The strong responsiveness of Env38 to SVCV infection suggested its important role in antiviral immunity of zebrafish. To provide support for this notion, we examined the cellular distribution of Env38 on leukocytes in SVCV-stimulated zebrafish. The cells expressed with Env38 (Env38$^+$ cells) were prepared from leukocytes through FCM sorting, and the phenotypic characteristics of the Env38$^+$ cells were examined on the basis of the hallmark genes expression of various immune cells. The results showed that the Env38$^+$ cells expressed CD80, membrane IgM (mIgM) and MHC-IIα, but did not express TCR-α and TCR-β, as shown in the reverse transcription PCR (RT-PCR) results (Fig 5B) and double immunofluorescence staining (Figs 4, S6A and S6B). FCM analysis showed that the percentage of Env38$^+$ cells in lymphatic and myeloid leukocytes in SVCV-stimulated zebrafish (21.70% ± 1.02%) dramatically increased (p < 0.01) compared with that of PBS-treated fish (2.13% ± 0.05%). The percentage of MHC-IIα and Env38 double positive cells (MHC-II$^+$Env38$^+$) significantly increased (p < 0.01) from 0.42% ± 0.02% to 18.60% ± 0.35% (Fig 5C). These results

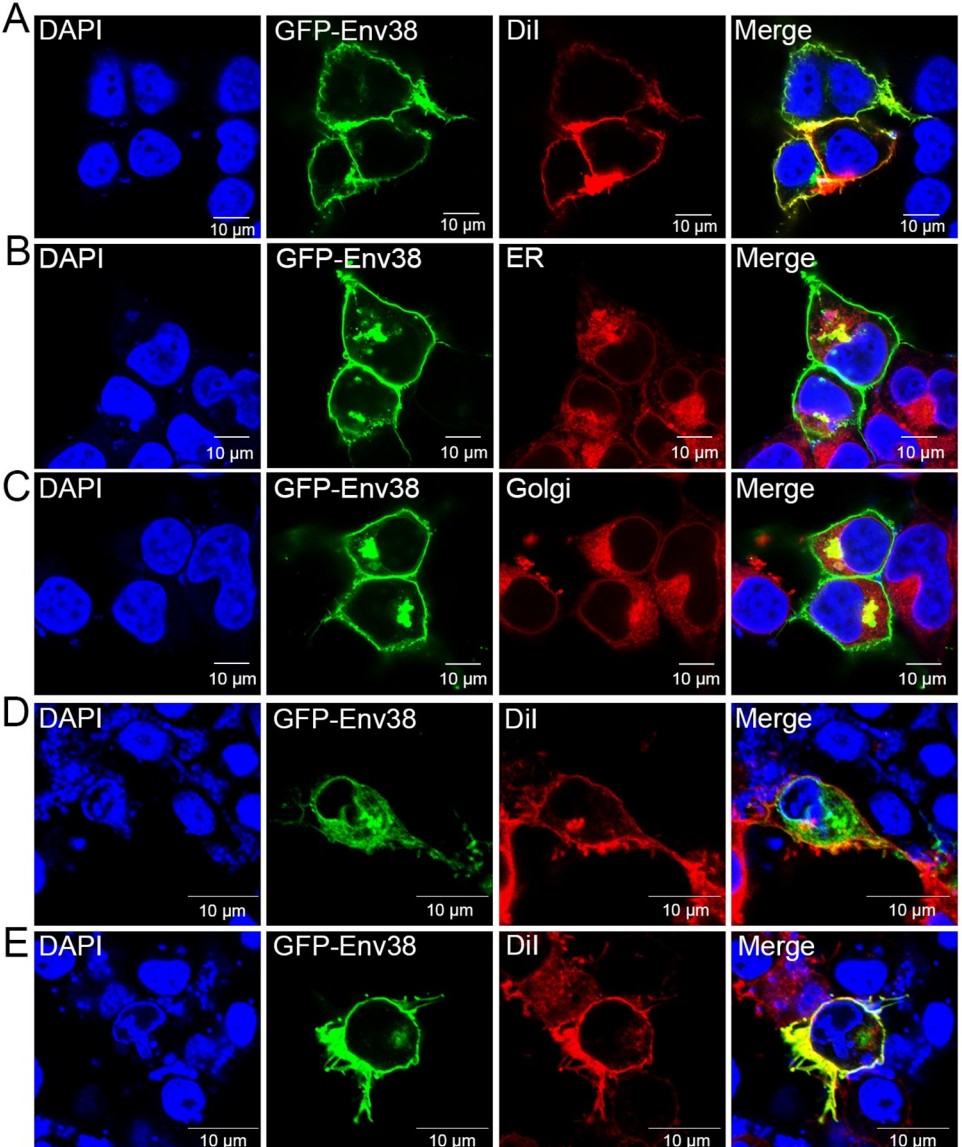

**Fig 3. Subcellular localization analysis of Env38 in HEK293T and EPC cells.** (A-C) HEK293T cells were transfected with the recombinant expression plasmid of pEGFPN1-Env38 (0.6 μg/ml) for 48 h, and then fixed and stained with the cell membrane probe DiI (A), ER-tracker (B) or Golgi-tracker (C) and nuclei probe DAPI. (D and E) EPC cells were transfected with the recombinant expression plasmid of pEGFPN1-Env38 (0.6 μg/ml) for 24 h and stimulated with SVCV (100 $TCID_{50}$) (E) or not (D) for another 24 h, and then fixed and stained with the cell membrane probe DiI and nuclei probe DAPI. The blue, green, and red fluorescence showed DAPI-labeled nuclei, Env38–EGFP fusion protein, and DiI–labeled cell membrane or ER-tracker–labeled ER or Golgi-tracker–labeled Golgi apparatus. Fluorescence images were captured using Laser scanning confocal microscope (FV3000) with 60 × oil glass.

revealed that Env38 was expressed on MHC-II[+] APCs (including IgM[+] B cells) upon inducing with SVCV, indicating the important role of Env38 in SVCV-induced adaptive immune response.

## Involvement of Env38 in SVCV-induced adaptive immunity

To evaluate the functional role of Env38 in host antiviral immune defense, an *in vivo* protective activity of Env38 against SVCV infection was examined in zebrafish through Ab-mediated blockade and siRNA-mediated knockdown assays. Anti-Env38 Ab (10 μg/fish) or

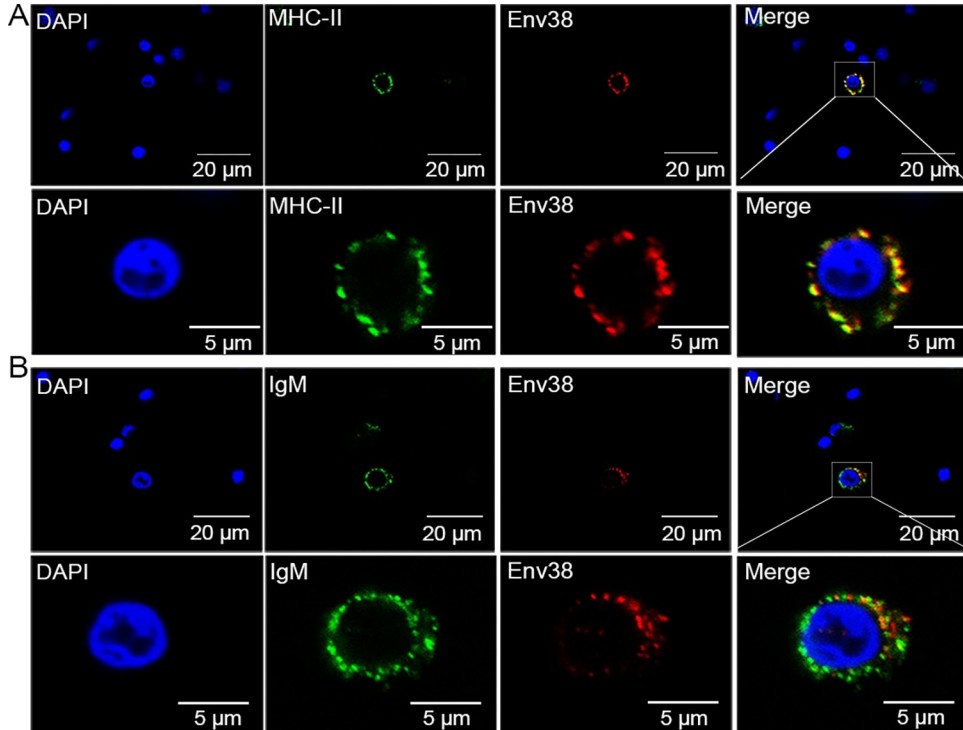

**Fig 4. Double immunofluorescence staining of MHC-II⁺Env38⁺ APCs in leukocytes of zebrafish.** The leukocytes were sorted from spleen, head kidney and peripheral blood of zebrafish stimulated with SVCV ($10^5$ TCID$_{50}$). (A) Cells were labled with mouse anti-Env38 and rabbit anti-MHC-IIα Abs, followed by Alexa Fluor 594-conjugated goat anti-mouse IgG and FITC-conjugated goat anti-rabbit IgG. (B) Cells were labled with mouse anti-Env38 and rabbit anti-IgM Abs, followed by Alexa Fluor 594-conjugated goat anti-mouse IgG and FITC-conjugated goat anti-rabbit IgG. DAPI stain showed the location of the nuclei. Fluorescence images were captured using a Laser scanning confocal microscope (FV3000) with 60 × oil glass.

Env38siRNA-LV ($1 \times 10^6$ TU/fish) was injected intraperitoneally (i.p.) three times into zebrafish infected with SVCV ($10^5$ TCID$_{50}$). The number of dying fish suffered from SVCV infection was recorded continuously for 21 days, and the survival rate for each group was calculated. The results showed that the survival rate of zebrafish was significantly declined in groups with blockade or knockdown of Env38 compared with that in control groups without the intervention of Env38 (Fig 6A). In this case, the survival rate of zebrafish markedly decreased from 53.33% ± 1.58% (SVCV plus nonrelated isotype IgG) and 47.80% ± 0.90% (SVCV plus scrambled contsiRNA-LV) in the control groups to 17.80% ± 0.90% (SVCV plus anti-Env38) and 27.80% ± 0.90% (SVCV plus Env38siRNA-LV) in Env38 blockade and knockdown groups (p < 0.05). Notably, adaptive humoral immunity against SVCV infection was markedly impaired by the blockade or knockdown of Env38, as shown by the significantly decline (p < 0.05) in the production of SVCV-specific antibody (IgM) in the serum of fish administered with anti-Env38 Ab and Env38siRNA-LV under challenge of SVCV after 7, 14, and 21 days (Fig 6B). These results indicated the important role of Env38 in host antiviral immunity, which was closely associated with SVCV-induced adaptive humoral immune response.

## Regulatory role of Env38 in CD4⁺ T cell activation

The cellular distribution of Env38 on MHC-II⁺ APCs suggested the potential role of Env38 in APC-initiated Ag-specific CD4⁺ T cell activation. Thus, this function was further examined

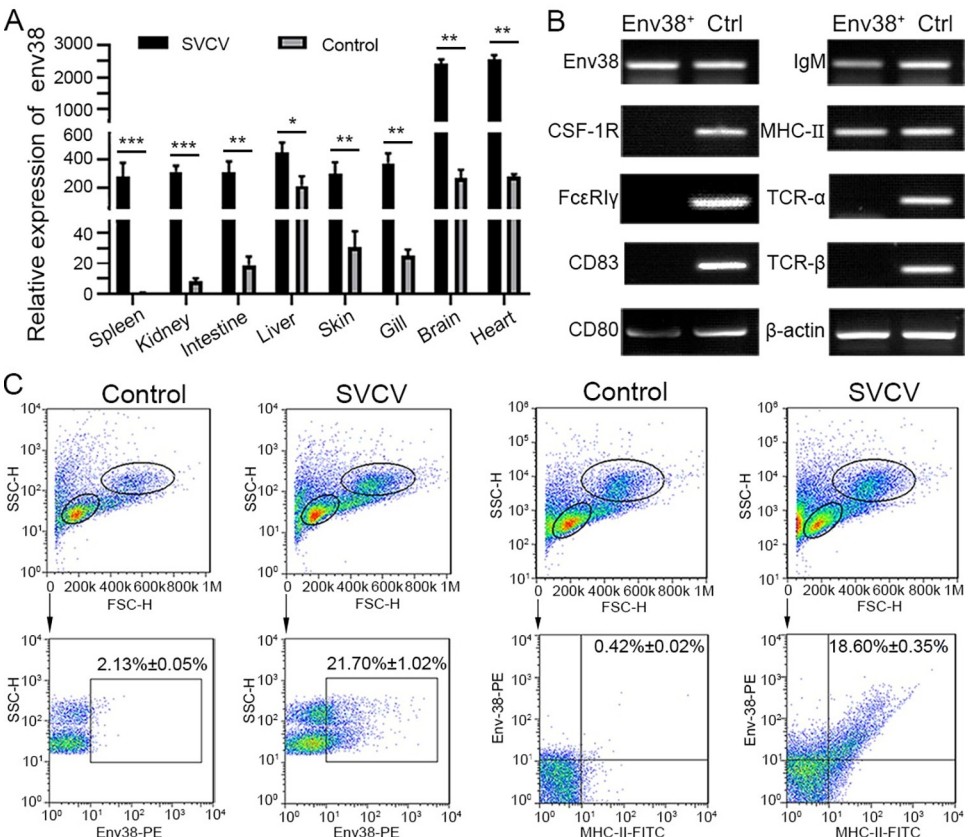

**Fig 5. Detection of Env38 in tissues and cells in response to SVCV infection.** (A) RT-qPCR analysis for the expression of *env38* gene in spleen, head kidney, intestine, liver, skin, gill, brain and heart tissues of zebrafish stimulated with SVCV ($10^5$ TCID$_{50}$) or mock PBS (control) for 7 days. The expression of *env38* mRNA in spleen in PBS treatment was arbitrarily set at 1. Negative control group was treated with mock PBS. Each sample was obtained from at least 10 fish. (B) Cellular distribution of Env38 on leukocytes from spleen, head kidney and peripheral blood of zebrafish stimulated with SVCV ($10^5$ TCID$_{50}$). The Env38$^+$ APCs were sorted from the leukocytes by FCM and qualitatively examined by RT-PCR with a panel of cellular markers of various immune cells, including CSF-1R and FcεRIγ (markers of monocytes/macrophages), CD83 and CD80 (markers of dendritic cells), membrane IgM (markers of B lymphocytes), MHC-II (markers of antigen presenting cells) as well as TCR-α and TCR-β (markers of T lymphocytes). Total leukocyte sample was used as a control. (C) Flow cytometry analysis of the percentages of Env38$^+$ cells and MHC-II$^+$Env38$^+$ cells in lymphatic and myeloid leukocytes from spleen, head kidney and peripheral blood of zebrafish stimulated with SVCV ($10^5$ TCID$_{50}$) for 7 days. The numbers above the outlined areas indicated the percentage of cells in each group. In control groups, zebrafish were i.p. with the same amount of mock PBS. In the RT-qPCR assay, PCRs were run in combination with the endogenous β-actin control. Error bars represent SEM. ($^*p < 0.05$; $^{**}p < 0.01$; $^{***}p < 0.001$; ns, no significant difference).

with *in vitro* and *in vivo* blockade and knockdown assays. For *in vitro* assay, the sorted MHC-II$^+$ APCs were stimulated with inactivated SVCV (100 TCID$_{50}$) and cocultured with the CFSE-labeled cognate responder CD4$^+$ T$_{svcv}$ cells (antigen/SVCV-specific CD4$^+$ T cells) sorted from leukocytes of zebrafish vaccinated with inactivated SVCV ($10^6$ TCID$_{50}$), during which mouse anti-Env38 Ab (5 μg/ml) and isotype mouse IgG (5 μg/ml; nonrelated control) were added into the cocultures. The proliferation of CD4$^+$ T$_{svcv}$ cells were examined through FCM analysis based on the dilution of CFSE and the upregulation of *lck* and *cd154* genes through RT-qPCR. The results showed that the proliferation of CD4$^+$ T$_{svcv}$ cells in response to SVCV-loaded APCs was significantly declined (p < 0.05) in Env38-blockade groups (19.63% ± 1.15%) compared with that in isotype IgG-treated groups (35.83% ± 1.21%). In this case, the percentage of CD4$^+$ T cells in Env38-blockade groups tends to approach that of the negative

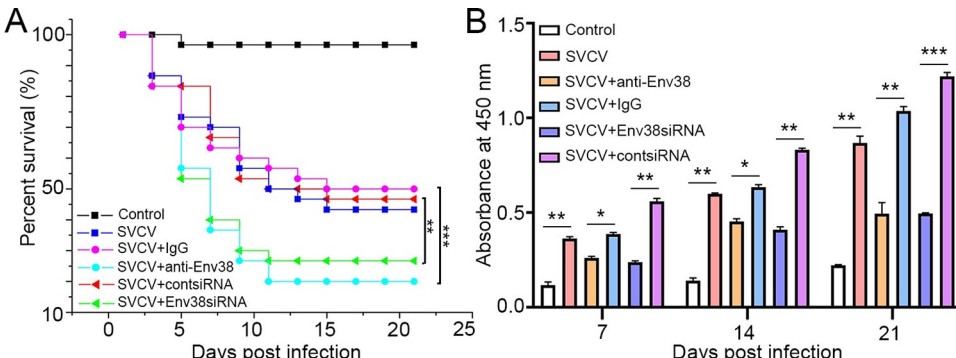

**Fig 6. Evaluation on the functional role of Env38 in host antiviral immune defense and IgM antibody production.**
(A) Survival rate was calculated in anti-Env38 Ab-mediated blockade and Env38siRNA-LV-mediated knockdown treatment in zebrafish injected with SVCV ($10^5$ TCID$_{50}$) in continuous 21 days. Kaplan-Meier survival curves represented data pooled from three independent experiments. Statistical differences were analyzed by log-rank test. 30 zebrafish were used in each experiment (n = 30). (B) ELISA for kinetic production analysis of the anti-SVCV antibody (IgM) in serum of zebrafish with treatments of Ab-mediated blockade and siRNA-mediated knockdown of Env38 under challenge of SVCV ($10^5$ TCID$_{50}$) after 7, 14 and 21 days. Error bars represented SEM. Isotype IgG-treated groups and scrambled siRNA-LV treated groups served as nonrelated control groups (n = 30). Negative control groups received PBS treatment. Error bars represented SEM. ($^*p < 0.05$; $^{**}p < 0.01$; $^{***}p < 0.001$; ns, no significant difference).

control groups (10.76% ± 0.78%), in which MHC-II$^+$ APCs were treated with mock PBS (Fig 7A). Meantime, the expression levels of *cd154* and *lck* in the blockade co-cultures were significantly downregulated (Fig 7B). For *in vivo* assay, zebrafish were i.p. administered with anti-Env38 Ab (10 μg/fish) or Env38siRNA-LV ($1 \times 10^6$ TU/fish) under SVCV stimulation. Expectedly, the activation of CD4$^+$ T cells was significantly inhibited in Env38-blockade and Env38-knockdown groups compared with that of the control groups administered with nonrelated isotype IgG or scrambled siRNA-LV. In this case, the proportion of the CD4$^+$CD154$^+$ T cells remarkably declined (p < 0.05) from 14.20% ± 1.07% (SVCV plus isotype IgG) and 12.30% ± 0.26% (SVCV plus contsiRNA-LV) in the control groups to 5.99% ± 0.73% (SVCV plus anti-Env38) and 6.15% ± 0.42% (SVCV plus Env38siRNA-LV) in the Env38 blockade and knockdown groups (Fig 7C). By contrast, the minimal change of the CD4$^+$CD154$^+$ T cells was observed among SVCV-stimulated (11.20% ± 0.14%), isotype IgG-treated (14.20% ± 1.07%), and contsiRNA-LV-treated (12.30% ± 0.26%) control groups. The expression levels of *cd154* and *lck* were remarkably downregulated (p < 0.05) in the Env38 blockade and knockdown groups (Fig 7D). In addition, the proliferative response of CD4$^+$ T cells to the blockade and knockdown of Env38 was assessed by *in vivo* staining of CD4$^+$ T cells with fluorescent EdU [33]. FCM analysis showed an overall decrease in the proliferation rate of the CD4$^+$ T cells in blockade and knockdown groups compared with that in the control groups (Fig 7E). The proportion of the CD4$^+$EdU$^+$ T cells significantly declined (p < 0.01) from 38.54% ± 1.78% (SVCV plus isotype IgG) to 6.47% ± 0.36% (SVCV plus anti-Env38) in the blockade assay and from 39.20% ± 2.42% (SVCV plus contsiRNA-LV) to 6.29% ± 0.58% (SVCV plus Env38siR-NA-LV) in the knockdown assay. By contrast, no significant change of CD4$^+$EdU$^+$ T cells was observed among SVCV-stimulated (38.27% ± 1.52%), isotype IgG-treated (38.54% ± 1.78%), and contsiRNA-treated (39.20% ± 2.42%) control groups. All these results provided *in vitro* and *in vivo* experimental evidence that Env38 was important for SVCV-induced CD4$^+$ T cell activation, suggesting the costimulatory role of Env38 in APC-initiated CD4$^+$ T cell response in adaptive humoral immunity.

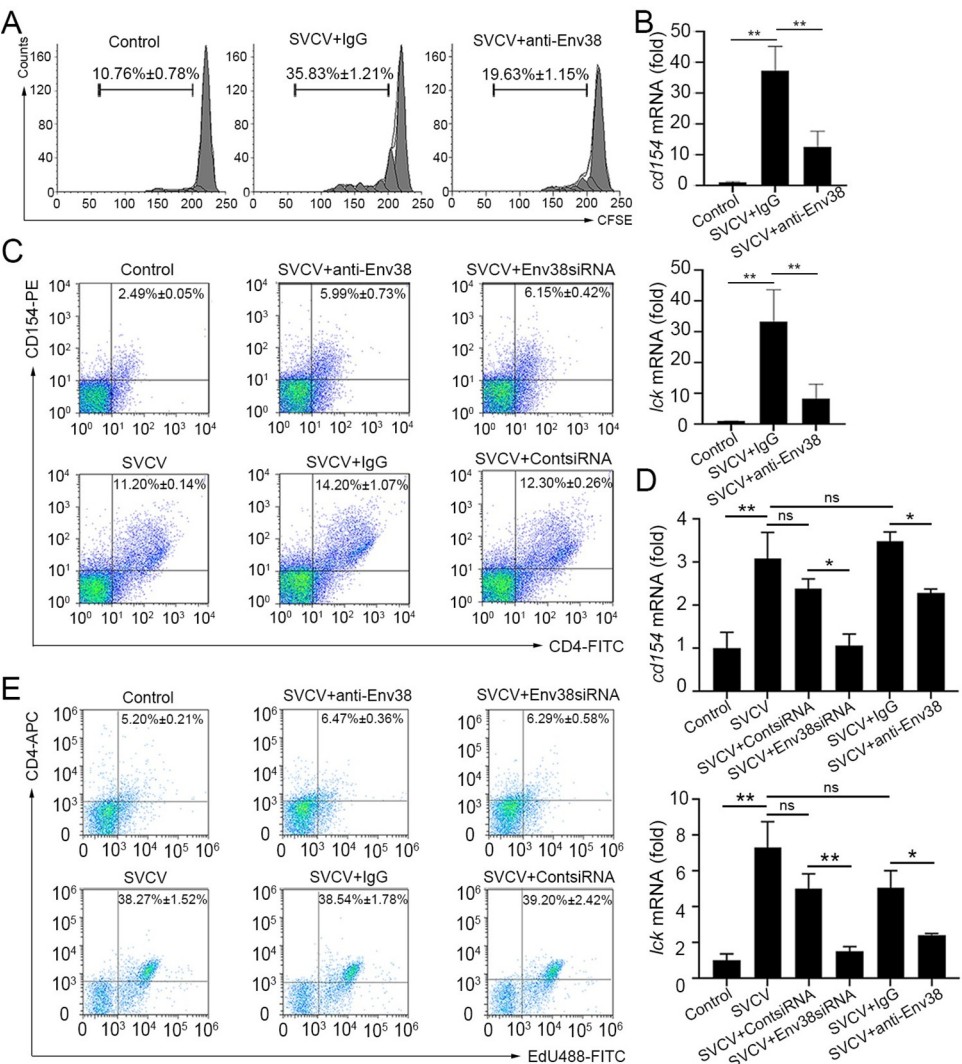

**Fig 7. Evaluation on the functional role of Env38 in APC-initiated Ag-specific CD4+ T cell activation.** (A and B) *In vitro* examination on the proliferation (A) and activation (B) of CD4+ $T_{SVCV}$ cells isolated from zebrafish stimulated with inactivated SVCV ($10^6$ $TCID_{50}$) and impaired by blockade of Env38 on APCs with anti-Env38 Ab (5 μg/ml) as determined by the CFSE dilution through FCM analysis and the transcriptional expression of *cd154* and *lck* genes through RT-qPCR. (C and D) *In vivo* examination on the activation of CD4+ T cells stimulated with SVCV ($10^5$ $TCID_{50}$) and impaired by blockade or knockdown of Env38 with i.p administering anti-Env38 Ab (10 μg/fish) or Env38siRNA-LV ($1×10^6$ TU/fish) as determined by the percentage of CD4+CD154+ T cells with mouse anti-CD154 (1:500) and rabbit anti-CD4 (1:500) in leukocytes from spleen, head kidney and peripheral blood through FCM analysis (C) and the transcriptional expression of *cd154* and *lck* genes through RT-qPCR (D). (E) *In vivo* examination on the proliferation of CD4+ T cells of zebrafish stimulated with SVCV ($10^5$ $TCID_{50}$) and impaired by blockade or knockdown of Env38 with i.p administering anti-Env38 Ab (10 μg/fish) or Env38siRNA-LV ($1×10^6$ TU/fish) as determined by the percentage of CD4+EdU+ T cells with rabbit anti-CD4 (1:500) in leukocytes from spleen, head kidney and peripheral blood as determined by FCM analysis. Nonrelated control groups were administered with isotype IgG or scrambled siRNA-LV. Negative control groups received mock PBS. The data presented in each block diagram indicated the average percentage of CD4+CD154+ T cells or CD4+EdU+ T cells in each treatment group. RT-qPCRs were run in combination with the endogenous β-actin control. Error bars represented SEM. ($^*p < 0.05$; $^{**}p < 0.01$; $^{***}p < 0.001$; ns, no significant difference).

## Effects of Env38 on B cell activation and Ab production

To further address the regulation of Env38 in adaptive humoral immunity, the involvement of Env38 in CD4+ T cell-initiated B-cell activation and Ab (IgM/IgZ) production were examined through *in vivo* Ab-mediated blockade and siRNA-mediated knockdown assays. As expected, upon stimulation with SVCV, the percentage of IgM+CD40+ B cells in the blockade or knockdown groups significantly declined ($p < 0.05$) from 23.30% ± 1.78% (SVCV plus isotype IgG) to 4.38% ± 0.87% (SVCV plus anti-Env38) in the blockade assay and from 24.90% ± 1.64% (SVCV plus contsiRNA-LV) to 4.94% ± 0.43% (SVCV plus Env38siRNA-LV) in the knockdown assay. By contrast, the percentage of IgM+CD40+ B cells was comparable among the isotype IgG-treated (23.30% ± 1.78%), contsiRNA-treated (24.90% ± 1.64%), and SVCV-stimulated (21.40% ± 1.16%) control groups (Fig 8A). Correspondingly, the transcriptional expression levels of *IgM* and *cd40* genes and the production of IgM Ab were remarkably downregulated ($p < 0.05$) in Env38 blockade and knockdown groups (Figs 8B and 6B). Similarly, the percentage of IgZ+ B cells significantly declined ($p < 0.05$) in the blockade and knockdown groups. The proportion of IgZ+ B cells declined from 14.00% ± 0.16% (SVCV plus isotype IgG) to 8.37% ± 0.41% (SVCV plus anti-Env38) or from 14.20% ± 0.83% (SVCV plus contsiRNA-LV) to 8.78% ± 0.49% (SVCV plus Env38siRNA-LV) in the leukocytes from spleen, head kidney, and peripheral blood (Fig 8C) and from 13.70% ± 0.86% (SVCV plus isotype IgG) to 4.87% ± 0.09% (SVCV plus anti-Env38) or from 13.50% ± 0.44% (SVCV plus contsiRNA-LV) to 6.50% ± 0.52% (SVCV plus Env38siRNA-LV) in the leukocytes from the gill (Fig 8D). The percentage of IgZ+ B cells in normal SVCV-infected group was similar to those in isotype IgG-treated and scrambled siRNA-treated control groups (Fig 8C and 8D). Accordingly, the production of *IgZ* mRNA and IgZ Ab decreased ($p < 0.01$) in Env38 blockade and knockdown groups compared with that in the control groups (Fig 8E and 8F).

## Functional verification of Env38 in knockout zebrafish

Finally, the functional role of Env38 in adaptive humoral immunity was verified by the impairment of the antiviral activity of Env38-/- zebrafish after vaccination with formaldehyde-inactivated SVCV vaccine. The survival rate of vaccinated fish significantly increased ($p < 0.01$) upon challenge with virulent SVCV ($10^5$ TCID$_{50}$) compared with that of unimmunized fish, suggesting that the adaptive antiviral immunity was well established in fish after vaccination (Fig 9A). However, the survival rate of vaccinated Env38-/- zebrafish significantly declined ($p < 0.05$) compared with that of immunized wild-type fish, suggesting that the deletion of Env38 attenuated vaccination-elicited immunoprotection against SVCV infection (Fig 9A). Notably, the survival rate also significantly reduced in non-vaccinated Env38-/- zebrafish, indicating that deletion of Env38 does not only attenuate protection upon immunization. It suggested that Env38 also plays an important role in the primary antiviral immune response of non-immunized zebrafish upon SVCV infection. Correspondingly, the viral loads of SVCV were greater in Env38-/- zebrafish compared with that in wildtype zebrafish (Fig 9B). The percentage of CD4+CD154+ T cells declined from 12.00% ± 0.73% in wild-type fish to 2.22% ± 0.05% in Env38-/- fish infected with SVCV (Fig 9C), and the percentage of IgM+CD40+ B cells declined from 25.40% ± 1.36% to 3.75% ± 0.15% (Fig 9D). Furthermore, the transcriptional levels of *cd154*, *lck*, *cd40*, and *IgM* mRNA were significantly downregulated ($p < 0.05$) in the Env38-/- zebrafish (Fig 9E). These results verified the important role of Env38 in SVCV-stimulated CD4+ T cell activation and downstream adaptive antiviral immune responses.

## Mechanism of Env38 in APC-initiated CD4+ T cell activation

Given the colocalization of Env38 with MHC-II on the surface of APCs, protein–protein interaction between Env38 and MHC-IIα or MHC-IIβ was investigated with Co-IP assay. Env38

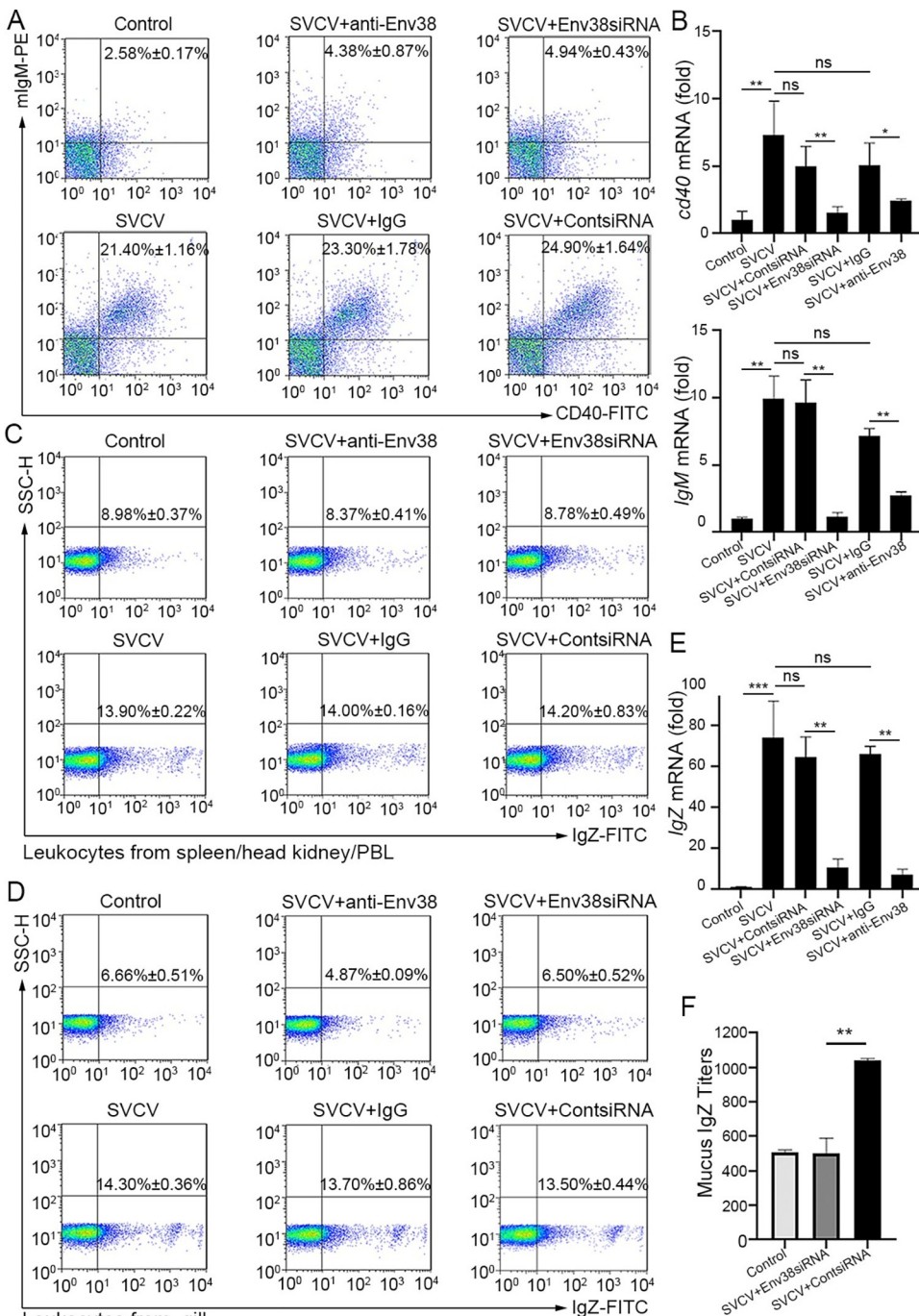

**Fig 8. Evaluation on the functional role of Env38 in CD4+ T cell-initiated B cell activation.** (A and B) *In vivo* examination on the activation of IgM+ B cells of zebrafish stimulated with SVCV ($10^5$ TCID$_{50}$) and impaired by blockade or knockdown of Env38 with i.p administering anti-Env38 Ab (10 μg/fish) or Env38siRNA-LV ($1\times10^6$ TU/ fish) as determined by the percentage of mIgM+CD40+ B cells with mouse anti-mIgM (1:2,000) and rabbit anti-CD40 (1:500) through FCM analysis (A) and the transcriptional expression levels of *cd40* and *IgM* genes through RT-qPCR (B). (C-F) *In vivo* examination on the activation of IgZ+ B cells of zebrafish stimulated with SVCV ($10^5$ TCID$_{50}$) and impaired by blockade or knockdown of Env38 with i.p administering anti-Env38 Ab (10 μg/fish) or Env38siRNA-LV ($1\times10^6$ TU/fish) as determined by the percentage of IgZ+ B with rabbit anti-IgZ (1:500) in leukocytes from spleen, head kidney and peripheral blood (C) and from gill (D) through FCM analysis, the transcriptional expression of *IgZ* genes via RT-qPCR (E) and the mucus IgZ antibody production by ELISA (F). Nonrelated control groups were administered with isotype IgG or scrambled siRNA-LV. Negative control groups received mock PBS. The data presented in each block diagram indicated the average percentage of IgM+CD40+B or IgZ+ B cells in each treatment group. RT-qPCRs

were run in combination with the endogenous β-actin control. Error bars represented SEM. ($^*p < 0.05$; $^{**}p < 0.01$; $^{***}p < 0.001$; ns, no significant difference).

and MHC-IIα or MHC-IIβ proteins were coexpressed in HEK293T cells. The Co-IP results showed that Env38 was strongly associated with MHC-IIα but not with MHC-IIβ (Fig 10A and 10B). The molecular interaction between Env38 and MHC-IIα proteins was also detected in leukocytes of zebrafish with SVCV stimulation (Fig 10C). To determine whether the N-glycosylation of Env38 affected interaction, we coexpressed Env38 and MHC-IIα in HEK293T cells, followed by tunicamycin treatment. The Co-IP result showed that the interaction was almost unchanged when N-glycosylation was inhibited (Fig 10D), suggesting that the association of Env38 with MHC-IIα was independent of Env38 glycosylation. Given that CD4 acts as a crucial coreceptor in TCR-pMHC binding through its association with MHC-II, we speculated that Env38 was probably involved in the formation and/or stabilization of TCR-pMHC-CD4 complex by cross-linking CD4 molecule. To test this hypothesis, we analyzed the interaction of Env38 with CD4 through Co-IP by coexpressing them in HEK293T cells. As expected, Env38 clearly interacted with CD4 protein (Fig 10E). The interaction was also

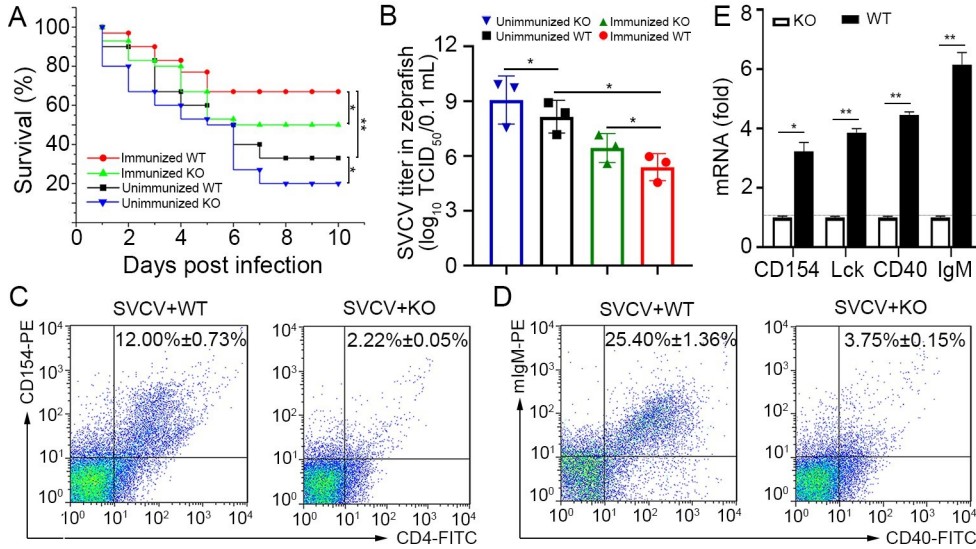

**Fig 9. Verification of the functional role of Env38 in knockout zebrafish.** (A) The functional role of Env38 was verified by the impairment of the antiviral activity of Env38$^{-/-}$ zebrafish after vaccination with formaldehyde-inactivated SVCV vaccine or infection with live SVCV. In both wildtype and knockout zebrafish, the immunized groups were i.p. inoculated with vaccine derived from 0.4% formaldehyde-inactivated SVCV ($10^6$ TCID$_{50}$) and then challenged with SVCV ($10^5$ TCID$_{50}$); whereas the unimmunized groups were challenged with SVCV ($10^5$ TCID$_{50}$) without vaccination. Data points were from three independent experiments, and 30 zebrafish were used in each experiment (n = 30). Differences were analyzed using log-rank test. (B) Viral titer (TCID$_{50}$) was measured to evaluate the viral load in the wild type and knockout zebrafish with or without vaccinated immunoprotection. Viral samples were prepared from zebrafish at 4th day post SVCV ($10^5$ TCID$_{50}$) infection. The TCID$_{50}$ was examined in EPC cells by using Reed-Muench method. Viral titers were log$_{10}$ transformed prior to statistical analysis. Data points were from three independent experiments, and three zebrafish were used for viral sample preparation in each experiment (n = 3). (C and D) The degree of CD4$^+$ T and IgM$^+$ B cell activation between WT and KO zebrafish under SVCV ($10^5$ TCID$_{50}$) stimulation was represented by the percentage of CD4$^+$CD154$^+$ T and IgM$^+$CD40$^+$ B cells with mouse anti-CD154 (1:500), rabbit anti-CD4 (1:500) (C) and mouse anti-mIgM (1:2,000), rabbit anti-CD40 (1:500) (D) through determination with FCM analysis. The data presented in each block diagram indicated the average percentage of CD4$^+$CD154$^+$ T and IgM$^+$CD40$^+$ B cells in WT and KO zebrafish. (E) The degree of CD4$^+$ T or IgM$^+$B cell activation between WT and KO zebrafish under SVCV ($10^5$ TCID$_{50}$) stimulation was represented by the transcriptional levels of *cd154*, *lck*, *cd40* and *IgM* through RT-qPCR. The RT-qPCRs were run in combination with the endogenous β-actin control. Error bars represented SEM. ($^*p < 0.05$; $^{**}p < 0.01$; $^{***}p < 0.001$; ns, no significant difference).

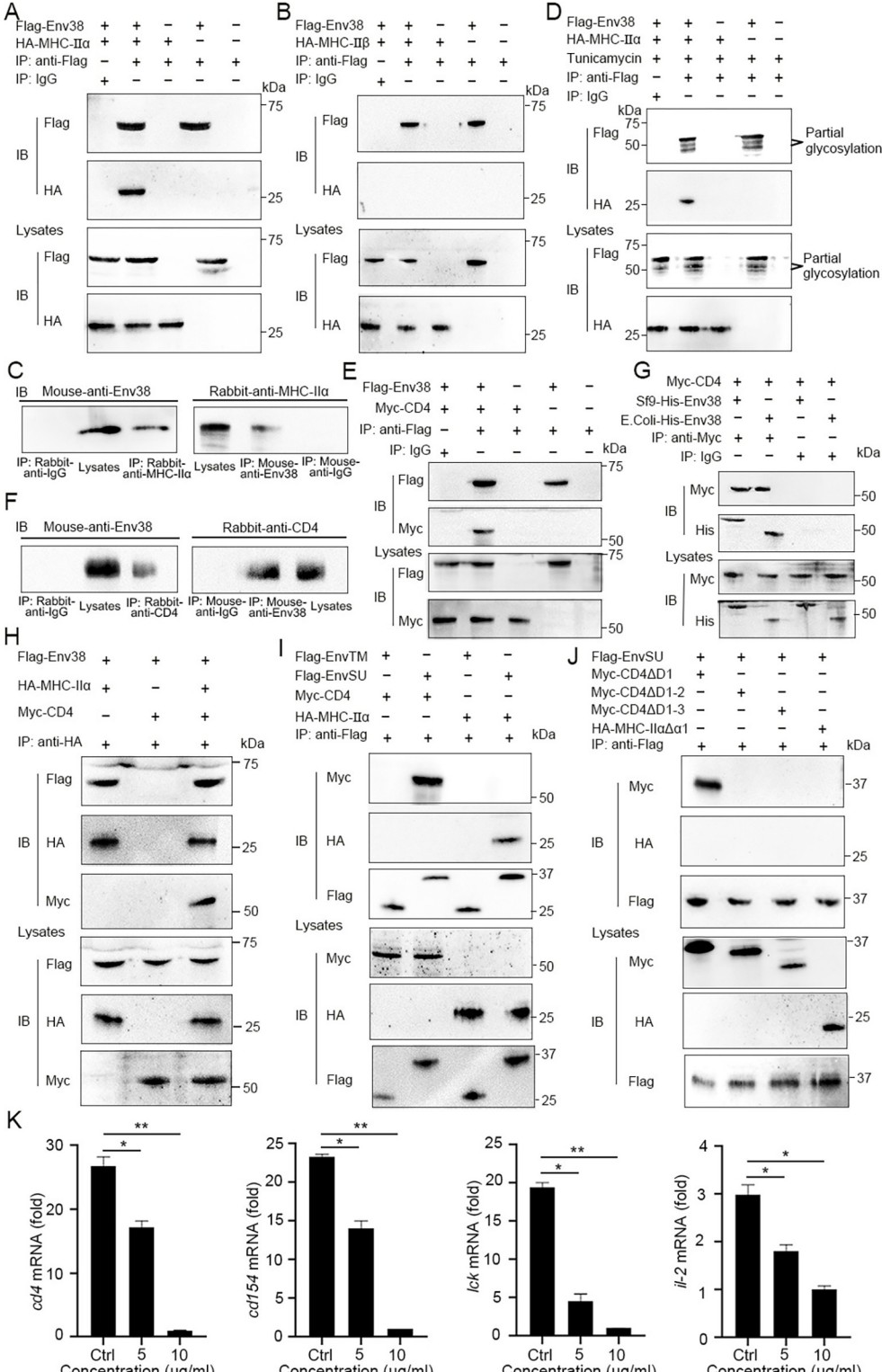

**Fig 10. Examination on the association of Env38 with MHC-II and CD4 proteins and their domain interactions.**
(A and B) Examination on the interaction between Env38 and MHC-IIα or MHC-IIβ, in which the Flag-tagged Env38 and HA-tagged MHC-IIα (A) or HA-tagged MHC-IIβ (B) were coexpressed in HEK293T cells, followed by immunoprecipitation with rabbit anti-Flag or rabbit IgG (negative control). Western blot was performed with mouse anti-Flag or anti-HA Ab. (C) Examination on the interaction between leukocyte Env38 and MHC-IIα proteins of

zebrafish under SVCV stimulation. The lysates were immunoprecipitated with rabbit anti-MHC-IIα or rabbit IgG. Western blot was performed with mouse anti-Env38 Ab. Similarly, the lysates were immunoprecipitated with mouse anti-Env38 or mouse IgG, and Western blot was performed with rabbit anti-MHC-IIα Ab. (D) Examination on the effect of glycosylation on the interaction between Env38 and MHC-IIα. The Flag-tagged Env38 and HA-tagged MHC-IIα were coexpressed in HEK293T cells under treatment with tunicamycin, followed by immunoprecipitation with rabbit anti-Flag or rabbit IgG. Western blot was performed with mouse anti-Flag or anti-HA Ab. (E) Examination on the interaction between Env38 and CD4, in which the Flag-tagged Env38 and the Myc-tagged CD4 were coexpressed in HEK293T cells, followed by immunoprecipitation with rabbit anti-Flag or rabbit IgG. Western blot was performed with the mouse anti-Flag or anti-Myc Ab. (F) Examination on the interaction between leukocyte Env38 and CD4 proteins of zebrafish under SVCV stimulation. The lysates were immunoprecipitated with rabbit anti-CD4 or rabbit IgG. Western blot was performed with mouse anti-Env38 Ab. Similarly, the lysates were immunoprecipitated with mouse anti-Env38 or mouse IgG, and Western blot was performed with rabbit anti-CD4 Ab. (G) Examination on the interactions between Env38 and CD4 proteins, in which the His-tagged Env38 proteins were purified from Sf9 or E. coli cells and the Myc-tagged CD4 was expressed in HEK293T cells, followed by immunoprecipitation with rabbit anti-Myc or rabbit IgG. Western blot analysis was performed with the mouse anti-Myc or anti-His Ab. (H) Examination on the interactions among Env38, MHC-IIα and CD4, in which the Flag-tagged Env38, HA-tagged MHC-IIα and Myc-tagged CD4 were coexpressed in HEK293T cells, followed by immunoprecipitation with rabbit anti-HA Ab. Western blot was performed with mouse anti-Flag, anti-HA or anti-Myc Ab. (I) Examination on the interactions among EnvTM, EnvSU, CD4 and MHC-IIα, in which the Flag-tagged EnvTM, Flag-tagged EnvSU, Myc-tagged CD4 and HA-tagged MHC-IIα were coexpressed in HEK293T cells in different combinations, followed by immunoprecipitation with rabbit anti-Flag Ab. Western blot was performed with mouse anti-Myc, anti-HA, or anti-Flag Ab. (J) Examination on the interactions among EnvSU, CD4ΔD1, CD4ΔD1-2, CD4ΔD1-3 and MHC-IIαΔα1, in which the Flag-tagged EnvSU, Myc-tagged CD4ΔD1, Myc-tagged CD4ΔD1-2, Myc-tagged CD4ΔD1-3 and HA-tagged MHC-IIαΔα1 were coexpressed in HEK293T cells in different combinations, followed by immunoprecipitation with rabbit anti-Flag Ab. Western blot was performed with mouse anti-Myc, anti-HA, or anti-Flag Ab. The same lysates were simultaneously immunoblotted to monitor the expression of CD4ΔD1, CD4ΔD1-2, CD4ΔD1-3, MHC-IIαΔα1, and EnvSU. In the Co-IP assays, the same lysates were simultaneously immunoblotted to monitor the expression of proteins in input. (K) Functional evaluation on the association among Env38, MHC-IIα and CD4 proteins distributed on MHC-II$^+$Env38$^+$APCs and CD4$^+$ T cells by examining the competitive inhibition of CD4$^+$ T cell activation by using a soluble D2 domain protein of CD4 (sCD4-D2, 5 μg/ml and 10 μg/ml). The activation of CD4$^+$ T cells was determined by the transcriptional expression levels of cd4, cd154, lck and il-2 genes through RT-qPCR. The RT-qPCRs were run in combination with the endogenous β-actin control. Error bars represented SEM. (*$p < 0.05$; **$p < 0.01$; ***$p < 0.001$; ns, no significant difference).

observed between Env38 and CD4 proteins naturally expressed on the spleen tissue of fish under SVCV stimulation (Fig 10F). Notably, Co-IP showed that CD4 interacted with the Env38 proteins from both prokaryotic and eukaryotic cells, suggesting that CD4 was associated with the protein framework rather than the modified glycans of Env38 (Fig 10G). Next, we confirmed the interactions among Env38, MHC-IIα and CD4 proteins through Co-IP by coexpressing them in HEK293T cells, which indicated the existence of a TCR-pMHC-CD4-Env38 complex in the immune synapse structure formed between MHC$^+$ APC and CD4$^+$ T cells during APC-initiated CD4$^+$ T cell activation (Fig 10H). To explore the mechanism underlying molecular interactions among Env38, MHC-IIα and CD4 proteins, a panel of truncated and mutant proteins with deletions of the various structural domains of Env38, MHC-IIα and CD4 were generated and coexpressed in HEK293T cells in different combinations for Co-IP assays. The truncated and mutant proteins included the SU subunit (1–226 amino acids; named EnvSU) and the TM subunit (231–427 amino acids; named EnvTM) of Env38, the mutant CD4 proteins with the deletions of the first Ig-like domain (124–476 amino acids; named CD4ΔD1), the first two Ig-like domains (212–476 amino acids; named CD4ΔD1-2), and the first three Ig-like domains (332–476 amino acids; named CD4ΔD1-3), and mutant MHC-IIα protein with deletion of α1 domain (102–236 amino acids; named MHC-IIαΔα1) (S8A and S8B Fig). The results showed that strong interactions existed between EnvSU and wild-type CD4 or CD4ΔD1, as well as EnvSU and wild-type MHC-IIα. However, no interactions were detected between EnvSU and CD4ΔD1-2 or CD4ΔD1-3, EnvSU and MHC-IIαΔα1, as well as EnvTM and wild-type MHC-IIα (Fig 10I and 10J). These outcomes demonstrated that the deletion of the SU subunit in Env38, the second Ig-like domain in CD4 (referred as

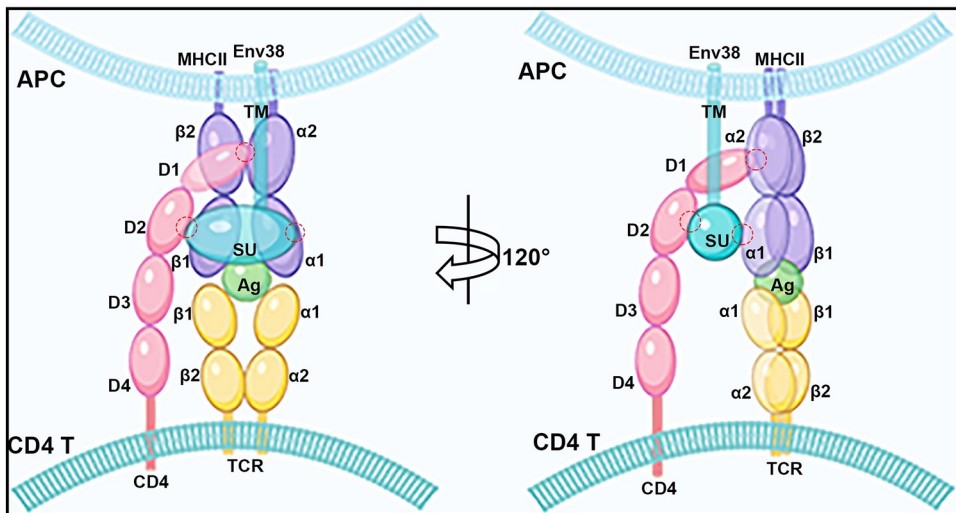

**Fig 11. A proposed model of the potential TCR-pMHC-II-CD4-Env38 complex.** The TCR-pMHC-II-CD4-Env38 complex formed in the immune synapse structure between MHC⁺Env38⁺ APC and CD4⁺ T cell during APC-initiated CD4⁺ T cell activation. Env38 located on the surface of MHCII⁺ APC in an integral form without undergoing cleavage between the transmembrane subunit (TM) and the surface subunit (SU). Env38 was associated with MHC-IIα and CD4 proteins, in which the SU subunit of Env38, the second immunoglobin domain of CD4 (CD4-D2) and the first α1 domain of MHC-IIα chain (MHC-IIα1) contribute to the association among Env38, CD4 and MHC-IIα proteins. In particular, Env38 specifically binds to CD4-D2 and MHC-IIα1 domains without intervention with the CD4-D1 and MHC-IIβ-β2 as well as CD4-D1 and MHC-IIα2 domains, two previously known reciprocal connectors contributing to the formation of the pMHC-TCR-CD4 complex referred as the central supramolecular activating complex (c-SMAC) in the immunological synapse. Hence, Env38 promoted the formation and/or stabilization of pMHC-TCR-CD4 complex via cross-linking MHC-II and CD4 molecules between the MHCII⁺ APCs and CD4⁺ T cells. The red circles represented the interaction domains among SU, CD4-D2 and MHC-IIα1.

CD4-D2), and the α1 domain in MHC-IIα chain inhibited molecular interactions between Env38 and CD4 or MHC-IIα, suggesting that the SU subunit, CD4-D2 and α1 domains largely contributed to Env38 interactions with CD4 and MHC-II proteins (Fig 11). Finally, we performed functional evaluation to support the evidence of the interactions among SU subunit, CD4-D2 and MHC-IIα1 domains in the proposed TCR-pMHC-CD4-Env38 complex. For this purpose, we prepared a recombinant soluble CD4-D2 domain protein (sCD4-D2) from the supernatants of transfected HEK293T cells by acetone precipitation and Ni-NTA Agarose affinity chromatography (S8C Fig). This sCD4-D2 was then introduced in the APC-initiated CD4⁺ T cell activation assay, in which sCD4-D2 was expected to attenuate CD4⁺ T cell activation by competitively binding to Env38, preventing membrane Env38 and CD4 association and disrupting TCR-pMHC-CD4-Env38 complex formation. Expectedly, the activation of CD4⁺ T cells was impaired by sCD4-D2, as determined by the observation that the expression levels of *cd4*, *cd154*, *lck* and *il-2* marker genes were suppressed by sCD4-D2 in a dose-dependent manner (Fig 10K).

### *env38* is implicated as an IFNφ1-stimulated gene

Previous studies have shown that SVCV infection in adult zebrafish induces a strong IFNφ response for IFNφ1 and IFNφ2 and a moderate response for IFNφ4 but shuts down the expression of IFNφ3 [34]. On the other hand, IFNs play an important role in antiviral immunity by upregulating numerous IFN-stimulated genes (ISGs) [35]. These observations suggest that *env38* may be an ISG potentially induced by IFNφ1 and/or IFNφ2. To explore this hypothesis, we produced zebrafish IFNφ1 and IFNφ2 proteins from the supernatants of transfected HEK293T cells by acetone precipitation and Ni-NTA Agarose affinity chromatography (Fig 12F). IFNφ1 and

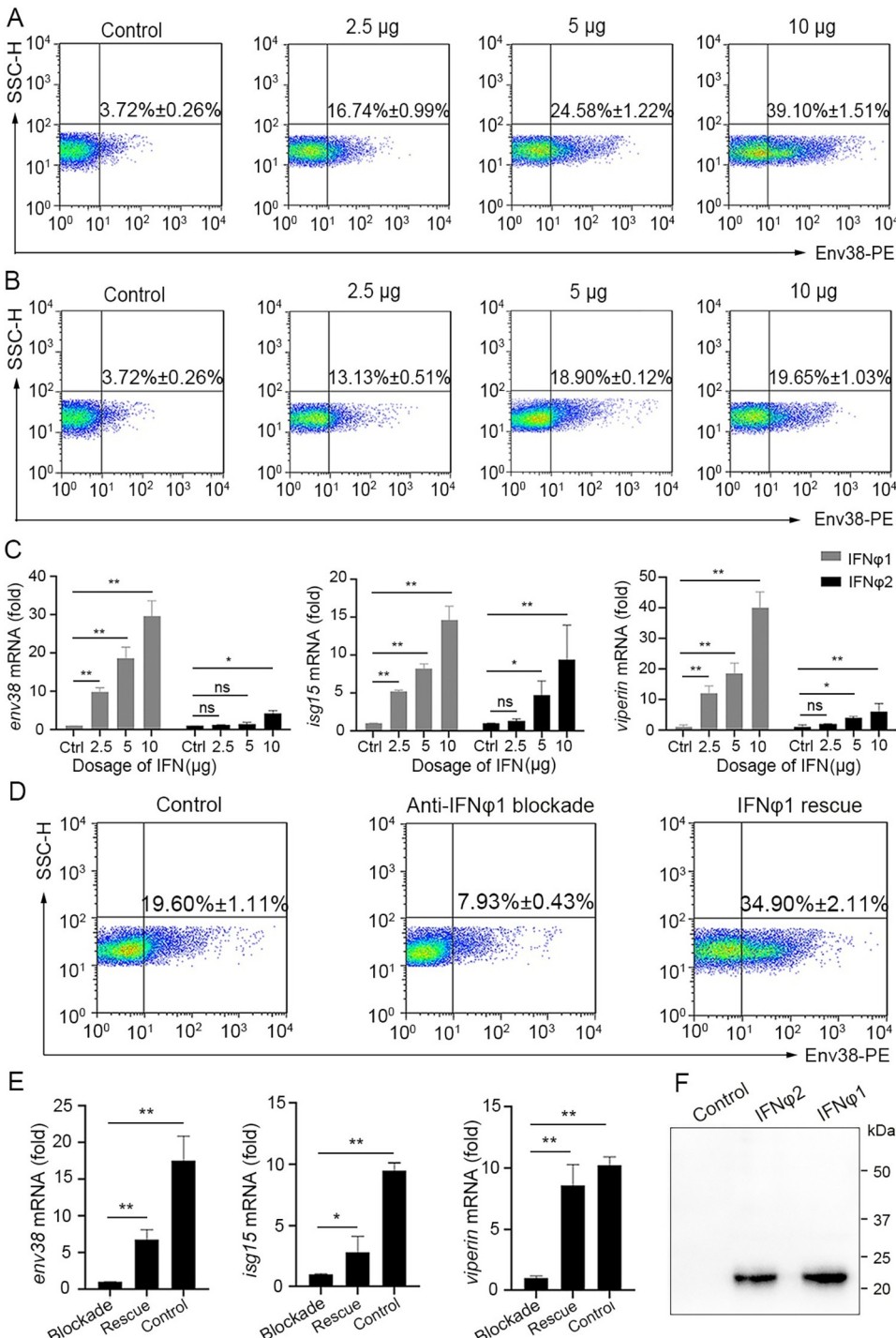

**Fig 12. Examination on the stimulatory role of zebrafish IFNφ1/IFNφ2 in Env38 production.** (A-C) Examination on the stimulatory role of zebrafish IFNφ1 and IFNφ2 in Env38 expression by *in vivo* administering recombinant IFNφ1 and IFNφ2 proteins. The stimulatory effect was examined by the increased percentage of Env38⁺ cells and the upregulated transcriptional expression of *env38*, *isg15* and *viperin* genes in leukocytes from spleen, head kidney and peripheral blood of zebrafish i.p. injected with proportional dosage (2.5 μg, 5 μg and 10 μg) of the IFNφ1 (A) or IFNφ2 (B) through FCM analysis (A and B) and RT-qPCR (C), respectively. Zebrafish i.p. injected with mock PBS was used as control. (D and E) Examination on the stimulatory role of zebrafish IFNφ1 in Env38 expression by *in vivo* neutralization and rescue assays via administering mouse anti-IFNφ1 Ab (10 μg/fish) and recombinant IFNφ1 (10 μg/ fish) back into the IFNφ1-neutralized zebrafish. The stimulatory effect was examined by the percentage of Env38⁺ cells

(D) and the expression level of *env38*, *isg15* and *viperin* genes (E) in leukocytes from spleen, head kidney and peripheral blood of zebrafish in each group under stimulation with SVCV ($10^5$ TCID$_{50}$). Zebrafish in the control groups were treated with isotype mouse IgG and rescued with mock PBS. (F) Western blot analysis of the recombinant IFNφ1 and IFNφ2 proteins with anti-Flag Ab (1:5,000) from supernatant of HEK293T cells transfected with the pcDNA3.1-His-Flag-IFNφ1 and pcDNA3.1-His-Flag-IFNφ2 recombinant constructs or an empty control construct. In the FCM analysis, different treatments were presented at the top of each block diagram, and the data presented in each block diagram indicated the average percentage of Env38$^+$ cells in each treatment group. RT-qPCRs were run in combination with the endogenous β-actin control. Error bars represented SEM. ($^*p < 0.05$; $^{**}p < 0.01$; $^{***}p < 0.001$; ns, no significant difference).

IFNφ2 proteins were administered intraperitoneally at different proportional dosages (2.5 μg, 5 μg and 10 μg) in the zebrafish. The induced expression of Env38 upon IFNφ1/IFNφ2 stimulation were determined by the proportional increase of Env38$^+$ cells in leukocytes and the transcriptional level of *env38* mRNA. FCM analysis results showed that the percentage of Env38$^+$ cells significantly increased (p < 0.05) from 16.74% ± 0.99% (2.5 μg) to 24.58% ± 1.22% (5 μg) and to 39.10% ± 1.51% (10 μg) with IFNφ1 dosage. The transcriptional level of *env38* mRNA also increased in the leukocytes of zebrafish stimulated with IFNφ1, consistent with the expression profiles of two typical ISGs (*isg15* and *viperin* genes) (Fig 12A and 12C). These findings indicated that *env38* was an ISG, which could be induced by IFNφ1 in a dose-dependent manner. In contrast, only slight changes in percentage of Env38$^+$ cells and transcriptional level of *env38* mRNA were observed in IFNφ2-treated zebrafish, in which the proportion of Env38$^+$ cells increased from 13.13% ± 0.51% (2.5 μg) to 18.90%± 0.12% (5 μg) and to 19.65% ± 1.03% (10 μg), respectively (Fig 12B and 12C). Obviously, IFNφ1 had a stronger inducing effect on *env38* expression than that of IFNφ2, suggesting that IFNφ1 played a major regulatory role in inducing *env38* expression. Next, we performed anti-IFNφ1 Ab-mediated blockade and IFNφ1 rescue assays in zebrafish challenged with SVCV ($10^5$ TCID$_{50}$) for further clarification. The results showed that the percentage of Env38$^+$ cells in leukocytes dramatically declined from 19.60% ± 1.11% in isotype IgG-treated control groups to 7.93% ± 0.43% in anti-IFNφ1 Ab-mediated blockade groups and this decline was significantly rescued by administering zebrafish with recombinant IFNφ1 protein (10 μg), as shown by the increase of Env38$^+$ cells from 7.93% ± 0.43% (blockade groups) to 34.90% ± 2.11% (rescue groups) (Fig 12D). Accordingly, the transcriptional levels of *env38*, *isg15* and *viperin* mRNAs were remarkably upregulated in the IFNφ1 rescue groups compared with those in the anti-IFNφ1 Ab-mediated blockade groups (Fig 12E). All the results above suggested that *env38* was an IFN-stimulated gene, and IFNφ1 acted as a major stimulator for Env38 production and functional regulation.

## Discussion

Despite their parasitic origins, an increasing number of investigations are finding ERV elements having been co-opted for beneficial immune function and regulation by the host cell [7]. It suggests that ERVs have turned out to represent important components of host immune system. Recent progress toward understanding the associations of ERV activity with immune activation include the discovery of Env glycoproteins restricting retroviral infection in primates [36], the identification of a SUMO-dependent pathway of innate antiviral immunity that involves the sensing of ERV nucleic acids [37], and the description of additional retroviral promoter and enhancer elements regulating antiviral gene expression [38–40]. These findings not only provide important insights into innate antiviral defense mechanisms but also help in assessing the role of ERV activation in infectious diseases. However, knowledge about the functions of ERVs in adaptive antiviral immunity remains limited. In the present study, we investigated the functional role of an ERV-derived envelop protein (Env38) in adaptive

humoral immunity against exogenous SVCV infection in a zebrafish model. Env38 is a typical glycosylated membrane protein mainly distributed on MHC-II$^+$ APCs. The strongly induced expression of Env38 in APCs upon SVCV infection indicated its role in APC-initiated adaptive antiviral immunity. In support to this notion, blockade and knockdown or knockout of Env38 dramatically impaired the activation of CD4$^+$ T cells, IgM$^+$ B and IgZ$^+$ B cell proliferation, IgM and IgZ Ab production, and zebrafish defense against SVCV challenge. These results provided experimental evidence of lines that Env38 plays an important stimulatory role in the initial activation of SVCV-induced CD4$^+$ T cells and the subsequent adaptive humoral immune responses. Mechanistically, Env38 was identified to be associated with MHC-IIα and CD4 proteins. It is well known that the CD4 is a crucial coreceptor of TCR on T cells and recognizes antigens displayed by an APC in MHC-II molecules, thus it is evident that Env38 helps CD4$^+$ T cell activation by promoting the formation and/or stabilization of pMHC-TCR-CD4 complex via cross-linking MHC-II and CD4 molecules between the APC and CD4$^+$ T cells. By constructing Env38, MHC-IIα and CD4 mutants with the deletions of various structural domains, the SU subunit of Env38, the second immunoglobin domain of CD4 (CD4-D2) and the first α1 domain of MHC-IIα chain (MHC-IIα1) were identified to contribute to facilitate interactions among Env38, CD4 and MHC-IIα proteins, in which the SU subunit of Env38 associates with CD4 and MHC-IIα by interacting with their CD4-D2 and MHC-IIα1 domains, respectively. Additionally, previous studies have shown that the first immunoglobin domain of CD4 (CD4-D1), the second β2 domain of MHC-IIβ chain (MHC-IIβ2), and the second α2 domain of MHC-IIα chain (MHC-IIα2) are responsible for the interaction between CD4 and MHC-II during the recognition of TCR to the antigens presented by the MHC-II complex [41,42]. Taken together, it is reasonable to understand that Env38 specifically binds to CD4-D2 and MHC-IIα1 domains without intervention with the CD4-D1 and MHC-IIβ domains, which constitute a pair of reciprocal connectors contributing to the formation of pMHC-TCR-CD4 complexes. However, the precise architecture of the pMHC-TCR-CD4-Env38 complex and the molecular interactions among the complex components remain to be further deciphered.

The primary viral function of Env glycoproteins is to facilitate entry into host cells, which involves binding to cell surface receptors and driving fusion of the virions and cellular membranes [43]. Generally, viral fusion proteins consist of three fusion protein classes (Class I-III), in which the Env glycoprotein of *Retroviridae*, the hemagglutinin (HA) glycoprotein of *Orthomyxoviridae*, the spike (S) glycoprotein of *Coronaviridae*, and glycoprotein (GP) of *Filoviridae* belong to Class I [26]. Fusion proteins in Class I are usually trimers in their pre-fusion and post-fusion states, and their final states typically have a central N-terminal trimeric α-helical coiled coil decorated by three C-terminal helices, thereby forming a six-helix bundle (6HB). Fusion peptides are at or near the N-terminus of the fusion subunit [26]. Class I fusion proteins are produced in a precursor form and then cleaved into TM and SU subunits by furin protease to liberate the N-terminal hydrophobic fusion peptide that would be inserted into the cell membrane [44,45]. When fusion commences, major structural rearrangements lead to the assembly of head-domain HR segments into a central trimeric α-helical coiled-coil structure, displacing the fusion peptide in the direction of the host cell membrane. Subsequent hairpin-like refolding then positions the heptad repeat (HR) domains into the grooves of the central triple helix, resulting in the formation of a 6HB post-fusion structure, in which the fusion peptide-proximal core coiled-coil structure is surrounded by three transmembrane domain (TMD)-proximal HR helices [46]. Zebrafish Env38 shares an overall coincident structure with class I fusion proteins. It contains a consensus furin protease cleavage site (R/K-X-R/K-R) between the SU and TM subunits. However, the natural Env38 protein in leukocytes, spleen and head kidney tissues was always kept integral, as indicated by Western blot analysis, suggesting that Env38 played a role as a precursor without undergoing enzymatic cleavage

process. The predicated partial hydrophobic fusion peptide indicated the inefficiency of Env38 in fusion function, which was consistent with the functional behavior of Env38 in promoting cell–cell adhesion rather than cell–cell fusion between the surface membranes of MHC-II$^+$ APCs and CD4$^+$ T cells. It suggested the functional exaptation of ERV-derived Envs through long-term purifying selection, in which the functionality of Env38 became different from that of Envs serving as a fusogen specialized in cell–cell fusion, such as Syncytins mediating the formation of a syncytium [10]. Notably, the non-cleavage behavior of Env38 can prevent the exposure of the immune suppressive domain from the N-terminal of TM, which ensures that Env38 exerts a positive stimulatory effect rather than a negative suppressive effect on CD4$^+$ T cell activation. The undetectable expression of FurinA and FurinB enzymes in the Env38$^+$ cells may help explain the non-cleavage reason of the Env38 precursor in APCs. However, other mechanisms that prevent furin proteases from cleaving at the existing furin cleavage site of Env38 were not excluded, which remain to be further explored in the follow-up study. In this respect, the knowledge of HIV, a modern exogenous retroviral model lineage, may provide valuable implications. It has known that a HIV infection is triggered by the binding of Env gp120 protein with approximately 24 N-linked oligosaccharides to CD4 molecules on the surface of target cells [47]. In HIV, glycans constitute approximately 50% of the molecular mass of gp120, and a large number of conserved epitopes on gp120 are shielded [48]. The non-glycosylated gp120 synthesized in the presence of tunicamycin fails to bind CD4, whereas the enzymatic removal of glycans from gp120 have no effect on CD4 binding. It suggests that the N-linked glycosylation is essential for the proper folding of gp120, rather than directly association with the CD4 protein [48,49]. In our study, a similar binding action occurs between CD4 and Env38 that contains about seven N-linked oligosaccharides, and the N-linked glycosylation of Env38 is unnecessary for its binding to CD4 proteins. Thus, we speculated that the N-linked glycosylation of Env38 is involved in the proper folding of Env38, which helps shield the furin cleavage site to a certain extent. Here, we emphasized the advantage of non-cleavage behaviour of Env38 in APCs. However, Env38 did retain the furin cleavage site instead of simply losing it, which suggested that Env38 may potentially involve in other physiological activities beyond immune reactions in a cleavage-dependent manner. Futher investigations are required to clarify this hypothesis.

Type I IFNs in teleost fish are categorized into two groups with distinct groups of receptors. IFNφ1 (a group I IFN) and IFNφ2 (a group II IFN) are strongly induced in zebrafish under SVCV infection [34]. Our present study showed that *env38* can be significantly induced in zebrafish through direct IFNφ1 exposure. In accordance with this result, two typical signal transducer and activator of transcription 1 (STAT1) binding sites were predicted in the 5′LTR and the 3′LTR regions of ERV-E5.1.38-DanRer, suggesting the involvement of the IFN/JAK/STAT signaling axis in the stimulation of *env38* expression. These findings indicated that ERV-E5.1.38-DanRer is an IFN-inducible ERV, and *env38* is an ISG, thereby adding a previously unrecognized *env* member to the ISG superfamily. Mounting evidence shows that ERVs act as a major source of regulatory activity in host immune responses by providing numerous ready-made modules of transcription factor binding sites and enhancer or promoter elements in LTRs [50]. The activities of ERV-derived enhancers and promoters are induced and tightly regulated by host innate immune signaling, particularly by the IFN signals [7]. It was found that ERVs were a sources of approximately 15% of binding sites for interferon-associated transcription factors STAT1 and IRF1 in human macrophages [38]. The primate-specific MER41 ERV family was identified as a major source of ERVs with interferon-inducible activity. For example, AIM2, an ISG acting as a sensor for cytoplasmic double-stranded DNA that regulates inflammasome assembly is regulated by MER41 [38,51]. These observations suggest that considerable ERVs are inducible enhancers that contribute to the regulation of ISGs. Thus, a

complex network regulation may exist between ERV and IFN systems, but their regulatory interactions and mechanisms in the network are not fully understood. It is now well-established that much of the regulation of ERVs and host immune responses is driven by epigenetic remodeling that control transcriptional changes during an infection [5]. The LTR elements are known as the foci of epigenetic silencing and activation. In this respect, the flanking LTRs of functional genes or *cis*-elements can undergo reprogramming through epigenetic modifications from repressive modules to active status [52]. Hence, the IFN-inducible ERV-E5.1.38--DanRer LTRs would become a valuable model platform for exploring epigenetic mechanisms underlying IFNφ1-stimulated *env38* expression under SVCV infection and will help enhance understanding of the complex network interactions among ERVs, IFNs, and epigenetic regulations during infection of an exogenous virus. In addition, RIG-I and MDA5 signaling pathways are involved in the recognition of SVCV infection and IFN response, enabling hosts to establish an antiviral state [53]. ERV-derived enhancers or promoters have since been shown to be directly activated by viral or bacterial infection [40]. Thus, further study is needed to elucidate whether ERV-E5.1.38-DanRer enhancers are directly activated by RIG-I/MDA5 signaling pathways under SVCV exposure; and whether a positive feedback loop regulation exits in *env38* expression, in which ERV-E5.1.38-DanRer enhancers are activated by RIG-I/MDA5 signaling, which promotes IFNφ1 production, thereby stimulated *env38* expression.

SVCV is a highly pathogenic aquatic rhabdovirus that induces contagious Spring viraemia of carp disease associated with dropsy and hemorrhagic symptoms in cyprinids and many other fish species [54]. Despite numerous studies on viral replication, histopathology, and host innate defense inducing type I IFNs to mitigate SVCV, knowledge about the molecular mechanisms of host adaptive immunity against SVCV infection remains limited [55]. In this study, we demonstrated the essential role of an ERV-derived Env38 protein in initiating CD4[+] T cell activation and downstream humoral immune responses against SVCV infection. The finding revealed a previously unknown regulatory mechanism of adaptive antiviral immunity against an exogenous virus by arousing an ERV activation. The interplay between ERVs and exogenous viruses is an interesting topic remaining to be explored. Although some studies have focused on the correlation between ERVs and XRVs, such as HIV [56–58], MuLV [15,59], and Jaagsiekte sheep retrovirus [60], the association of ERVs with non-retroviral viruses remains largely unclear. Therefore, our present study on the SVCV-induced Env38-associated adaptive antiviral immune response may become a helpful research model for understanding the relationship of ERVs with exogenous invading viruses and host immunity. The adaptive immunity provides long-term and highly specific host protection against pathogen infection. During the development of adaptive immunity, specific antigen peptides presented in association to the major histocompatibility complex (pMHC) by APCs are recognized by the T cell antigen receptor (TCR) on naïve T cells [61]. This event leads to the assembly of a specialized membrane domain at the cell–cell contact area between an APC and a T cell known as the immunological synapse, which is a key physical structure required to induce T cell activation, proliferation, and differentiation to antigen-specific helper T cells or effector cells to induce appropriate immune responses [62]. Protein components are segregated into two distinct regions within this contact area: a central area referred to as the central supramolecular activating complex (c-SMAC) and a peripheral region called the p-SMAC. The major components of the c-SMAC are TCR/pMHC complex and associated coreceptors (e.g., CD4 and CD8), costimulatory molecules (e.g., CD28/CD80/CD86) and signaling molecules (e.g., CD3 complex). The p-SMAC consists of cytoskeleton-related or adhesion molecules structurally supporting the immunological synapse, such as the leukocyte function-associated antigen-1/scaffolding protein talin (LFA-1/talin), intracellular adhesion molecule-1/LFA-1 (ICAM-1/LFA-1), CD2/LFA-3, CD2/CD48, or CD2/CD58 [63]. c-SMAC mediates antigen recognition and T cell

activation, whereas p-SMAC supports T cell–APC conjugation and maintains the architecture of the immunological synapse [64]. Although numerous proteins have been identified from the immunological synapse, whether other components exist in this synapse and how they function in T cell activation remain issues of great interest. Here, we found that Env38 acts as an intrinsic regulator in the central part of c-SMAC, which stimulates CD4$^+$ T cell activation possibly by organizing or stabilizing the interaction between TCR and pMHC via cross-linking MHC-II and CD4 molecules (Fig 11). To the best of our knowledge, this study is the first to identify an ERV-derived envelope protein that is a component in c-SMAC and plays an essential role in initiating adaptive antiviral immunity. This finding will greatly enrich current knowledge on the molecular mechanism underlying the activation of adaptive immune response.

## Materials and methods

### Ethics statement

All animal care and experimental procedures were approved by the Committee on Animal Care and Use and the Committee on the Ethic of Animal Experiments of Zhejiang University.

### Experimental fish

Wild-type AB zebrafish (*Danio rerio*) with body weights of 0.5–1.0 g and lengths of 3–4 cm was raised and maintained in recirculating water at 28˚C on a 12 h/12 h light/dark cycle under standard laboratory conditions as previously described [31]. All fish used in experiments were siblings generated after at least two generations of inbreeding. Only healthy fish, as determined by general appearance and activity level, were used in the experiments.

### Cell lines and virus

Human embryonic kidney 293T (HEK293T) cells were cultured in Dulbecco's modified Eagle's medium (DMEM, Gibco) supplemented with 10% (v/v) fetal bovine serum (FBS, Gibco) at 37˚C in 5% $CO_2$. Epithelioma papulosum cyprini (EPC) cells were cultured in Medium 199 (M199, Gibco) supplemented with 10% (v/v) FBS at 28˚C in 5% $CO_2$. Spodoptera frugiperda (Sf9) cells were cultured in SIM HF medium (Sino Biological) supplemented with 10% (v/v) FBS at 28˚C without $CO_2$. Spring viremia of carp virus (SVCV) was a gift from Prof. Yibing Zhang (Institute of Hydrobiology, Chinese Academy of Sciences). SVCV was propagated in EPC cells at 28˚C and titrated in 96-well plates until the cytopathic effect (CPE) was complete. SVCV was purified from EPC culture medium by sucrose gradient centrifugation at 100,000 g for 2 h as previously described [65], and finally suspended in sterile PBS and stored at -80˚C until use. Viral titer was determined to be $1.58 \times 10^7$ $TCID_{50}$/mL by 50% tissue culture infective dose ($TCID_{50}$) assay on EPC cells using the Reed-Müench method [66]. For vaccination purpose, SVCV was inactivated by formaldehyde with a final concentration of 0.4% (v/v) followed by the protocol as described [67].

### Bioinformatics analysis

The name of the ERV-E5.1.38-DanRer-Env (Env38 in brief) is derived from the nomenclature of ERV-E5.1.38-DanRer following the rules described previously [24]. Genome location of the *env38* gene was retrieved from the Genome Data Viewer in the NCBI database. The conserved structural domains and motifs in Env38 protein were identified by Conserved Domains Database (CDD) of NCBI. Primers for gene cloning were predicted by the PrimerBLAST program. Putative extracellular, transmembrane, and cytoplasmic regions were predicted by TMHMM

Server (version 2.0). The architecture analysis of CD4 and MHC-IIα were conducted by SMART. The cDNA sequences and deduced amino acids were described using DNAMAN. The protein hydrophobicity analysis was conducted by ExPASy. The potential Env38 protein structure was predicted with I-TASSER [68]. The tertiary structural figures were reviewed and colored in VMD software.

### Molecular cloning

The *env38* gene was predicted in our previous study [24]. Total RNA was extracted from SVCV treated zebrafish spleen with TRIzol Reagent (Invitrogen). The cDNA was synthesized according to manufacturer's protocol of the PrimeScript II Reverse Transcriptase (Takara). The *env38* sequence was amplified with specific primers and nested primers in our previous research [24]. The cDNA was sequenced on a 3730 XL sequencer (Applied Biosystems), and compared with genomic sequence. The *mhc-iiα* (GeneID:368878), *mhc-iiβ* (GeneID:368615), *furina* (GeneID:566557), *furinb* (GeneID:566591) and *cd4* (GenBank: JQ855487.1) sequences were acquired from the National Center for Biotechnology Information database (NCBI). The cDNAs of these genes were cloned with primers shown in S1 Table and sequenced as described above.

### Plasmid constructions

The encoding sequence of *env38* together with the noncoding sequences of 5'-R, U5, U3, and 3'-R with self-promoter element, the full-length *env38* coding sequence, the SU and TM segments of *env38* sequences, *mhc-iiα*, *mhc-iiαΔα1*, *mhc-iiβ*, *furina*, *furinb*, *cd4*, *cd4Δd1*, *cd4Δd1-2*, *cd4Δd1-3*, and *cd4-d2* segment sequences were amplified by PCR with primers shown in S1 Table. After gel extraction and digestion, the sticky fragments were inserted pEGFPN1 (BD Biosciences) with enhanced GFP, pcDNA3.1 (Invitrogen) with His and Flag tags, pDisplay (Invitrogen) with HA tag, pCMV (Invitrogen) with Myc tag, pAcGHLTc (Invitrogen) with His tag, and pET28a (Invitrogen) with His tag by using ClonExpress II One Step Cloning Kit (Vazyme, C112). The resulting constructs were designated as pEGFPN1-Env38, pcDNA3.1--Flag-Env38-LTR, pcDNA3.1-Flag-Env38SU, pcDNA3.1-Flag-Env38TM, pDisplay-HA-MHC-IIα, pDisplay-HA-MHC-IIαΔα1, pDisplay-HA-MHC-IIβ, pCMV-Myc-CD4, pCMV-Myc-CD4ΔD1, pCMV-Myc-CD4ΔD1-2, pCMV-Myc-CD4ΔD1-3, pcDNA3.1-His-Flag-CD4-D2, pDisplay-HA-FurinA, pDisplay-HA-FurinB, pAcGHLTc-His-Env38 and pET28a-His-Env38. These constructs were used for the examination of Env38 subcellular localization and preparation of recombinant proteins in eukaryotic and prokaryotic cells. Constructs for transfection were prepared free of endotoxin by using the Endo-Free Plasmid Mini Kit II (Omega Bio-Tek). PcDNA3.1-His-Flag-IFNφ1 and pcDNA3.1-His-Flag-IFNφ2 constructs were kept in our lab as described [69].

### Preparation of recombinant proteins

For preparation of intact Env38 protein in prokaryotic cells, pET28a-His-Env38 was transformed into *E. coli* BL21 (DE3, TransGen Biotech) competent cells, cultured in Lysogeny broth (LB) medium containing kanamycin (50 mg/l, Sangon Biotech) at 37˚C with 200 rpm shaking, and induced by Isopropyl β-d-1-thiogalactopyranoside (IPTG, 0.5 mM, Sangon Biotech) at 16˚C for 20 h. The competent cells precipitation was treated with ultrasound, and then the supernatants or precipitation were collected for purification. For baculovirus expression, pAcGHLTc-His-Env38 and linearized baculovirus DNA (AB Vector) were co-transfected into Sf9 cells under the assistance of polyethylenimine (branched PEI; Sigma-Aldrich) in a T25 flask with SIM HF medium containing penicillin (100 U/ml) streptomycin (100 μg/ml). The

cells were cultured at 28°C for 5 d. The harvested precipitation was dissolved in lysing buffer (200 mM Tris-HCl, pH 8, 150 mM NaCl, 1% Nonidet P-40, 1 mM PMSF). For preparation of the second Ig-like domain of CD4 (sCD4-D2), IFNφ1 and IFNφ2 proteins, pcDNA3.1-His-Flag-CD4-D2, pcDNA3.1-His-Flag-IFNφ1 and pcDNA3.1-His-Flag-IFNφ2 plasmids were transfected into HEK293T cells. After culturing in DMEM medium with 10% FBS at 37°C for 24 h, the fresh DMEM medium without FBS substituted for next 48 h. The recombinant proteins with His tag were precipitated from the culture supernatants by supplementing iced acetone (1:4, v/v) for 12 h at -20°C, purified by nickel-nitrilotriacetic acid agarose (Ni-NTA Agarose) affinity chromatography (Qiagen), concentrated with the Protein Concentrator (Thermo scientific) following the manufacturer's instructions, and then detected by 12% SDS-PAGE and Western blot as described [69]. The protein concentration was surveyed by bicinchoninic acid (BCA) assay with Enhanced BCA Protein Assay Kit (Beyotime).

## Preparation of polyclonal and monoclonal Abs

The Env38 protein expressed by *E. coli* cells was used to prepare polyclonal antibody (Ab) against Env38 protein (anti-Env38). Six-week-old male Balb/c mice (∼20 g) from Laboratory Animal Center (Zhejiang) were immunized with the Env38 protein (20 mg) in Freund's complete adjuvant (CFA; Sigma-Aldrich) initially and the above immune process was repeated 3 days later. After 28 days, the mice were immunized again with the Env38 protein (20 mg) in Freund's incomplete adjuvant (IFA; Sigma-Aldrich). Seven days after the final immunization, serum samples were collected. The anti-Env38 Ab was affinity purified by Protein A agarose column (Thermo Fisher Scientific), and the titer was examined by ELISA. The validity and specificity of the anti-Env38 Ab were determined by Western blot analysis. The Abs against zebrafish MHC-IIα, CD4, CD154, CD40, mIgM, IgZ and IFNφ1, including rabbit anti-MHC-IIα, rabbit anti-CD4, mouse anti-CD154, rabbit anti-CD40, rabbit anti-mIgM, rabbit anti-IgZ and mouse anti-IFNφ1 were produced in our previous studies [32,69–71]. Monoclonal antibodies including mouse anti-mIgM mAb and mouse anti-MHC-IIα mAb were prepared with recombinant prokaryotic proteins by HuaBio Co. Ltd. (Hangzhou, China).

## Quantitative real-time PCR

The expression of *env38*, *lck*, *cd154*, *IgM*, *cd40*, *il-2*, *isg15*, and *viperin* mRNAs in zebrafish leukocytes or tissues were analyzed by quantitative real-time PCR (RT-qPCR) on CFX Connect Real-Time System (Bio-Rad). Briefly, the total RNA was extracted with TRIzol Reagent and cDNAs were synthesized by the PrimeScript II Reverse Transcriptase as described above. RT-qPCR experiments were performed in a total volume of 10 μl by using iTaq Universal SYBR Green Supermix (Bio-Rad). The reaction mixtures were incubated for 3 min at 95°C, followed by 40 cycles of 15 s at 95°C, 15 s at 58°C, and 20 s at 72°C. The relative expression levels were calculated using the $2^{-\Delta\Delta Ct}$ method with β-actin for normalization. Bio-Rad CFX Manager software was used for data processing. Each RT-qPCR trial was run in triplicate parallel reactions and repeated three times. The primers used in RT-qPCR are listed in S1 Table.

## Tissue distribution and subcellular localization analyses

For tissue distribution analysis of *env38* mRNA, spleen, head kidney, intestine, liver, skin, gill, brain and heart tissues were collected from zebrafish infected with SVCV ($10^5$ TCID$_{50}$) or treated with mock PBS (control) for 7 days. RT-qPCR was performed for the quantification of *env38* mRNA as described above. For subcellular localization assay, HEK293T or EPC cells were seeded into 6-wells plate (Corning) with cover glass and cultured in high-glucose DMEM or M199 in which 10% (v/v) FBS was added to allow growth until 50–60% confluence. The

cells in each well were transfected with 0.6 µg/ml of pEGFPN1-Env38 or pcDNA3.1-Flag-Env38-LTR combined with PEI reagent (3 µg/ml) following the manufacturer's protocol. At 48 h post-transfection, HEK293T cells transfected with pEGFPN1-Env38 were fixed by 4% (m/v) paraformaldehyde (PFA, Sigma-Aldrich) at 37˚C for 10 min, and stained with cell membrane probe DiI (1:200; Beyotime) or ER-Tracker Red (1: 2,000, Beyotime) at 37˚C for 10 min, or Golgi-Tracker Red (1:200, Beyotime) at 4˚C for 30 min. HEK293T cells transfected with pcDNA3.1-Flag-Env38-LTR were fixed by 4% PFA, blocked with 2% BSA, labeled with mouse anti-Env38 Ab (1:1:500) followed by FITC-conjugated goat anti-mouse IgG (1:1,000; Thermo Fisher) at 37˚C for 1 h, and stained with DiI, or ER-tracker Red or Golgi-tracker Red as described above. Then the cells were stained by the nuclei probe 4',6-diamidino-2-phenylindole (DAPI, 100 ng/ml; Sigma-Aldrich) at 37˚C for 5 min. For protein trafficking analysis, EPC cells were transfected with pEGFPN1-Env38 for 24 h, and then stimulated with SVCV (100 $TCID_{50}$) or not for another 24 h. Thereafter, the cells were fixed and stained as described above. Fluorescence images were captured using Laser scanning confocal microscope (FV3000) with 60 × oil glass.

## Glycosylation analysis

The glycosylation of Env38 protein was examined by periodic acid-Schiff (PAS) reaction, peptide N-glycosidase F (PNGase F) digestion and tunicamycin inhibition. For PAS reaction, Env38 proteins produced from Sf9 and *E. coli* cells were separated by 12% SDS-PAGE as described above. Then the SDS-PAGE gels were stained for carbohydrate with PAS using the Glycogen Periodic Acid-Schiff Staining Kit (Beyotime) according to the manufacturer's instruction. For PNGase F digestion, Env38 protein produced from Sf9 cells was treated in a denaturation buffer (0.5% SDS, 40 mM DTT) at 100˚C for 10 min. Then, the protein sample was digested with PNGase F (500 U/ml, YEASEN) in a buffer containing 50 mM sodium phosphate and 1% NP-40 (pH 7.5). Each reaction was performed in 100 µl at 37˚C for 3 h. The deglycosylation was examined by the change of the molecular weight of Env38 protein after enzymatic digestion in 12% SDS-PAGE gel, and the blot was probed with the anti-His Ab (1:5,000, Invitrogen). For tunicamycin inhibition assay, HEK293T cells were transfected with pcDNA3.1-Flag-Env38-LTR (0.6 µg/ml) and treated with tunicamycin (5 µg/ml, Beyotime) for 48 h at 37˚C before protein extraction. The samples were analyzed by 12% SDS-PAGE, and the blot was probed with the anti-Flag Ab (1:5,000, Invitrogen). Image Lab 6.0 was used to analyze the molecular weight of Env38 protein with linear semi-log analysis method.

## Preparation of leukocytes

The spleen and head kidney tissues, and peripheral blood samples were collected from zebrafish with designated treatments in ice-cold D-Hank's with heparin sodium (10 U/ml, Sigma-Aldrich). Whole-blood cell suspensions were obtained with heparinized capillary tubes, and single-cell suspensions of spleens and head kidneys were prepared by gently teasing the tissues through an 80-mm nylon mesh filter. Leukocytes were enriched from the cell suspensions by Ficoll-Hypaque (1.080 g/ml) centrifuging at 2,500 rpm for 25 min at room temperature, collected from the interface layer, and washed thrice with ice-cold D-Hank's at 350 g for 10 min at 4˚C. For primary cell culture, the isolated leukocytes or sorted lymphocytes were seeded into 24-wells plate (Corning) with Leibovitz's L-15 Medium (L-15, Gibco) including 10% (v/v) FBS, 100 U/ml of penicillin and 100 µg/ml of streptomycin at 28˚C in 5% $CO_2$ for indicated time. Cell viability was assayed with Trypan Blue (Trypan Blue Staining Cell Viability Assay Kit, Beyotime) and the leukocyte samples with cell viability ≥ 95% were used in the next experiments.

## Flow cytometry analysis and sorting

The isolated leukocytes were blocked with 2% bovine serum albumin (BSA, Sigma-Aldrich) for 1 h at 4°C and incubated with the defined primary Abs, including mouse anti-Env38 (1:500), rabbit anti-MHC-IIα (1:500), mouse anti-CD154 (1:500), rabbit anti-CD4 (1:500), mouse anti-mIgM (1:2,000), rabbit anti-CD40 (1:500) and rabbit anti-IgZ (1:500) for 2 h at 4°C. Nonspecific rabbit or mouse IgG (1:2,000, Thermo Fisher) was served as the nonrelated control. After washing thrice with D-Hank's buffer, the cells were incubated with secondary Abs (PE-conjugated goat anti-mouse IgG, 1:500 and FITC-conjugated goat anti-rabbit IgG, 1:1,000, Thermo Fisher) for 1 h at 4°C. The cells were detected by analytical FCM (FACSCalibur, BD Biosciences) and sorted by FCM (FACSJazz, BD Biosciences). FCM analysis for Env38$^+$ cell, MHC-II$^+$Env38$^+$ cell, CD4$^+$CD154$^+$ cell, and CD40$^+$IgM$^+$ cell and FCM sorting for MHC-II$^+$ cell, CD4$^+$ cell, and Env38$^+$ cell were performed following previously described protocols [31]. At least 10,000 cells were acquired from the gate for analysis. FlowJo 7.6 software (BD Biosciences) was used for data processing.

## Immunofluorescence staining

The colocalization of Env38 and MHC-IIα, or Env38 and mIgM was determined by immunofluorescence staining. Leukocytes were isolated as described above. The cells were fixed by 4% PFA at room temperature for 10 min. For intracellular protein staining, the fixed cells were permeabilized with 0.1% Saponin (Sigma) in PBS at room temperature for 10 min. Next, the cells were blocked with 2% BSA, and incubated with primary Abs in combinations of mouse anti-Env38 (1:500) and rabbit anti-MHC-IIα (1:500), as well as mouse anti-Env38 (1:500) and rabbit anti-IgM (1:500) at 4°C for 2 h. After washing with D-Hank's, the cells were combined with secondary Abs including Alexa Fluor 594-conjugated goat anti-mouse IgG (1:1,000, Thermo Fisher) and FITC-conjugated goat anti-rabbit IgG (1:1,000), following the manufacturer's instructions. The cells were washed with D-Hank's again and stained with DAPI (100 ng/ml) at 37°C for 5 min. Fluorescence images were captured using Laser scanning confocal microscope (FV3000) with 60 × oil glass.

## Co-IP assay for association of Env38 with MHC-IIα and CD4

For co-immunoprecipitation (Co-IP) assay, HEK293T cells were co-transfected with pcDNA3.1-Flag-Env38-LTR (0.3 μg/ml) and pDisplay-HA-MHC-IIα (0.3 μg/ml) or pDisplay-HA-MHC-IIβ (0.3 μg/ml) or pCMV-Myc-CD4 (0.3 μg/ml); pcDNA3.1-Flag-Env38SU (0.3 μg/ml) and pCMV-Myc-CD4 (0.3 μg/ml) or pDisplay-HA-MHC-IIα (0.3 μg/ml); pcDNA3.1-Flag-Env38TM (0.3 μg/ml) and pCMV-Myc-CD4 (0.3 μg/ml) or pDisplay-HA-MHC-IIα (0.3 μg/ml); pcDNA3.1-Flag-Env38SU (0.3 μg/ml) and pCMV-Myc-CD4ΔD1 (0.3 μg/ml) or pCMV-Myc-CD4ΔD1-2 (0.3 μg/ml) or pCMV-Myc-CD4ΔD1-3 (0.3 μg/ml) in a 10-cm dish under the assistance of PEI (3 μg/ml), and collected at 48 h post transfection. Besides, leukocytes were isolated from spleen, head kidney and peripheral blood of zebrafish stimulated with SVCV ($10^5$ TCID$_{50}$). Then the cells were lysed with precooling cell lysis buffer (Beyotime) containing protease inhibitor mixture (Roche) for 30 min at 4°C. The lysates were centrifuged (12,000 g at 4°C for 10 min), and the supernatants were incubated with defined primary antibodies at 4°C overnight. The mixture was incubated with 50 μl of protein A agarose beads (Thermo Fisher Scientific) for 4 h. The protein-beads complex was washed three times with lysis buffer, mixed with loading buffer, and then heated in 100°C for 10 min to denature the proteins. After centrifugation (12,000 g at 4°C for 10 min), the proteins were separated by 12% SDS-PAGE and transferred onto a PVDF transfer membranes (Millipore Sigma) for Western blot analysis. After blocking with 2% BSA at 4°C for 1 h, the membrane was incubated with

specific primary antibodies at 4˚C for 2 h, washed with TBST for thrice, and then incubated with HRP-conjugated goat anti-mouse/rabbit IgG mAb (1:8,000, Abmart) at 4˚C for 1 h. Expression of the transfected constructs was also analyzed in the whole cell lysates as input controls. The supernatants incubated with rabbit or mouse anti-IgG (1:5,000) were administered as controls. In addition, the supernatant of HEK293T cells transfected with pCMV-Myc-CD4 (0.6 μg/ml) was incubated with Env38 proteins (5 μg/ml) prepared from *E. coli* and Sf9 cells at 4˚C overnight, followed by immunoprecipitation with anti-Myc Ab (1:5,000, Invitrogen) and Western blot analysis as described above. The objective proteins were visualized with ECL reagents (GE Healthcare) by a digital gel image system (Tanon 4500).

## Enzyme-linked immunosorbent assay (ELISA)

The levels of serum IgM and mucus IgZ Abs against SVCV were measured by an indirect ELISA. The serum and skin/gill mucus samples were prepared from zebrafish as previously described [71]. The polystyrene microtiter wells were coated with SVCV (100 μl/well) in coating buffer (15 mM $Na_2CO_3$, 35 mM $NaHCO_3$, pH 9.6) overnight at 4˚C. The wells were washed thrice with PBST (PBS with 0.05% Tween-20) and then blocked with 200 μl of 3% BSA in PBS at 37˚C for 1 h. After washing, the serum (diluted at1:10) or mucus (diluted at different ratios) was added (200 μl/well) and incubated at 37˚C for 2 h. Following washing, mouse anti-IgM mAb or rabbit anti-IgZ Ab (10 μg/ml, 200 μl) was added to each well. After incubation at 37˚C for 1 h, the wells were washed for three times with PBST for 5 min, and incubated with 200 μl of goat-anti-mouse or goat-anti-rabbit Ig-HRP conjugate (1:8,000) for 1 h. The wells were washed for four times with PBST for 5 min. Then, 100 μl of 3,3′,5,5′-tetramethylbenzidine solution (TMB, Tiangen) was added to each well for color development at room temperature in the dark for 20 min. The reaction was stopped by adding 50 μl of 2 M $H_2SO_4$. The pre-immunization serum or mucus were used as negative controls. The absorbance was measured on a Synergy H1 Hybrid Multi-Mode Microplate Reader (BioTek) at 450 nm. The Ab level was determined by absorbance at 450 nm (for serum IgM) or titer (for mucus IgZ), the latter of which is defined as the highest mucus dilution at which the $A_{450}$ ratio ($A_{450}$ of post-immunization / $A_{450}$ of pre-immunization) is greater than 2.1 [72].

## Generation of short hairpin RNA-encoding lentivirus

The short hairpin RNAs (shRNAs) carrying the small interfering RNAs (siRNAs) targeting *env38* mRNA or scrambled siRNA (control) and the shRNA-encoding lentiviruses (LVs) were designed and produced as previously described [31]. The shRNAs were constructed into a pSUPER construct (pSUPER.retro.puro, OligoEngine) downstream of the H1 promoter. The constructs were transfected into HEK293T cells with pEGFPN1-Env38 for efficiency evaluation with RT-qPCR. The U6 promoter cassette in a lentiviral construct (pLB) was replaced by an H1-shRNA cassette from the construct harboring the most effective shRNA to produce a pLB-siRNAEnv38 vector. The shRNA-encoding LV was generated by cotransfecting HEK293T cells with pLB-siRNAEnv38 and packaging vectors (pCMV-VSVG and pCMV-dR8.2). The viral supernatant was concentrated by ultracentrifugation (25,000 rpm, 90 min, 4˚C) and resuspended in PBS. To quantify the infectious lentiviral vector titer by FCM analysis, adherent HEK293T cells were infected with serially diluted LV samples and the expression of the EGFP fusion protein was detected using Zeiss Axiovert 40 CFL with 100 × original magnification. The functional lentiviral titer, given in transducing units (TU) per ml, was calculated using the following formula: TU/ml = (initial cell count × % GFP positive/inoculation volume)/dilution [73,74]. The silencing activity of the resulting LV was determined in zebrafish leukocytes by RT-qPCR and FCM analysis or spleen tissues by Western blot analysis after

fish were i.p. injected with the LV ($1 \times 10^6$ TU/fish) once every 24 h for three times with SVCV stimulation ($10^5$ TCID$_{50}$).

### Generation of Env38 knockout zebrafish

The sgRNAs for *env38* were designed by Chop-Chop [75]. The DNA templates for *in vitro* sgRNA synthesis, which contained the T7 promoter, a 20-nucleotide (nt) long guide sequence, and an sgRNA scaffold [76], were generated by overlapping PCR. The sgRNA-F sequence is ATAATACGACTCACTATAGAATCGTCACCAGG GGCCCACAGTTTTAGAGCTAGAA ATAGC, and the sgRNA-R sequence is AAAAGCACCGACTCGGTGCCACTTTTTTCAAG TTGATAACGGACTAGCCTTATTTTAACTGCTATTTCTAGCTCTAAAAC. The sgRNA transcription *in vitro* was carried out in reaction containing 1 μg of DNA template, 2 μl of reaction buffer, 2 μl of ATP (100 mM), 2 μl of GTP (100 mM), 2 μl of UTP (100 mM), 2 μl of CTP (100 mM), 2 μl of T7 RNA Polymerase Mix (HiScribe T7 High Yield RNA Synthesis Kit, NEB), and water up to 20 μl. Transcription was performed by incubation at 37˚C for more than 16 h, followed by treatment with RNase-free DNase I (NEB) at room temperature for 30 min. The synthesized sgRNAs were purified as the protocol of MEGAclear Kit (Invitrogen, AM1908). Cas9 protein (500 ng/μl, ThermoFisher, A45220P) and gRNA (90 ng/μl) were co-injected into one-cell-stage zebrafish embryos, and embryos were raised at 28˚C in E3 medium (5 mM NaCl, 0.17 mM KCl, 0.33 mM CaCl$_2$ 0.33 mM MgSO$_4$, 0.1% methylene blue) and then transferred to the standard circulating system. For identification of Env38 knockout, the caudal fins of one month old zebrafish were lysed in the lysis solution (1ml TE buffer, 5 μl Tween20, 5 μl proteinase K). The settings for lysis included 65˚C for 30 min, during which vortex several times, 100˚C for 8 min, and then centrifuge in 13,000 g for 2 min. Take the supernatant as test sample, genotyping was identified with the specific primers shown in S1 Table by PCR. The successfully knocked out fish were raised individually and mated for homozygous zebrafish.

### *In vitro* assay for Env38 on CD4$^+$ T cell activation

The primary MHC-II$^+$ APCs were sorted from the leukocytes of unstimulated zebrafish with mouse anti-MHC-IIα mAb by FCM sorting as described above, then the cells were stimulated with 0.4% formaldehyde-inactivated SVCV (100 TCID$_{50}$) at 28˚C for 24 h, and washed with L-15 medium (Gibco) three times to remove noningested SVCV, during which the MHC-II$^+$ APCs received mock PBS treatment were served as negative control. Meantime, the antigen-specific CD4$^+$ T cells (CD4$^+$ T$_{svcv}$) sorted by FCM with rabbit anti-CD4 from the leukocytes of zebrafish vaccinated by injection with inactivated SVCV ($10^6$ TCID$_{50}$) for 7 d, were stained with preheated 5 μM CFSE (Beyotime) at room temperature for 20 min. The reaction was terminated by supplementing with 10% FBS in cold L-15 medium and incubated on ice for 5 min. After being washed thrice with the L-15 medium, the CFSE-labeled responder CD4$^+$ T cells were cocultured with the MHC-II$^+$ APCs in L-15 medium with 10% FBS, 100 U/ml of penicillin and 100 μg/ml of streptomycin at 28˚C for 3 d, during which mouse anti-Env38 Ab (5 μg/ml), isotype mouse IgG (5 μg/ml; nonrelated control) and soluble CD4-D2 domain protein (sCD4-D2, 5 μg/ml and 10 μg/ml) were added into the cocultures, respectively. The proliferation of CD4$^+$ T$_{svcv}$ cells were examined through FCM analysis based on the dilution of CFSE and analyzed by ModFit LT program. The activation of CD4$^+$ T$_{svcv}$ cells was also determined by the upregulation of *lck*, *cd154*, *il-2*, and *cd4* genes through RT-qPCR with primers shown in S1 Table.

### *In vivo* assay for Env38 on CD4$^+$ T and B cell activation and Ab production

*In vivo* Ab-mediated blockade and Env38siRNA-LV-mediated knockdown assays were performed to evaluate the effects of Env38 on CD4$^+$ T cell activation and downstream B cell

activation and IgM/IgZ Ab production. For the knockdown assay, zebrafish stimulated with SVCV ($10^5$ TCID$_{50}$) were i.p. injected thrice with 10 µl Env38siRNA-LV ($1\times10^6$ TU/fish) at 24 h intervals [32]. Scrambled siRNA-encoding LV was administered as a nonrelated control. For the blockade assay, zebrafish were i.p. stimulated with SVCV ($10^5$ TCID$_{50}$) and administered thrice with mouse anti-Env38 Ab (10 µg/fish) at 24 h intervals. Isotype mouse IgG was injected (10 µg/fish) as a nonrelated control. In these cases, the fish injected with mock PBS was served as negative control group. At 7 days after the stimulation of SVCV, the activation of CD4$^+$ T and IgM$^+$ B cells was evaluated by the increased percentage of CD4$^+$CD154$^+$ T and IgM$^+$CD40$^+$ B cells in the leukocytes from spleen, head kidney and peripheral blood by FCM analysis with mouse anti-CD154 (1:500), rabbit anti-CD4 (1:500), mouse anti-mIgM (1:2,000), rabbit anti-CD40 (1:500) followed by PE-conjugated mouse anti-IgG (1:500) and FITC-conjugated rabbit anti-IgG secondary Abs (1:1,000), respectively. Additionally, the transcriptional expression of *lck*, *cd154*, *IgM* and *cd40* genes was examined by RT-qPCR. For examination of CD4$^+$ T cell proliferation, the fish under Env38 blockade/knockdown treatment were i.p. injected with 5-ethynyl-2'-deoxyuridine, Alexa Fluor 488 (EdU, Beyotime, 8 µg/fish) twice every 24 h two days before sacrifice. Seven days after the stimulation of SVCV, the leukocytes from spleen, head kidney and peripheral blood were collected and incubated with rabbit anti-CD4 (1:500) followed by APC-conjugated rabbit anti-IgG secondary Ab (1:1,000; Thermo Fisher). Then, cells were fixed with 4% PFA and permeabilized with 0.3% Triton X-100 in PBS, followed by click reaction according to the manufacturer's instruction (BeyoClick EdU Cell Proliferation Kit with Alexa Fluor 488, Beyotime). Thereafter, the proliferation of CD4$^+$ T cells were analyzed by FCM (CytoFLEX, BECKAMN). Meantime, the serum samples were collected from fish at 7, 14 and 21 days and mucus samples were collected from fish at 7 days after the stimulation of SVCV, and the level of IgM and IgZ Abs against SVCV were examined by ELISA as described above. Besides, the activation of CD4$^+$ T and IgM$^+$ B cells evaluated by the increased percentage of CD4$^+$CD154$^+$ T and IgM$^+$CD40$^+$ B cells by FCM analysis and the transcriptional levels of *lck*, *cd154*, *IgM* and *cd40* genes examined by RT-qPCR were also examined in Env38 knockout (Env$^{-/-}$) zebrafish.

## Immunoprotection assay

Immunoprotection assay was performed to further evaluate the functional role of Env38 in host antiviral immune defense with the immunized groups and unimmunized control group. One of the immunized groups was i.p. inoculated with the vaccine derived from 0.4% formaldehyde-inactivated SVCV ($10^6$ TCID$_{50}$) in wild type (WT) zebrafish. The other immunized group was inoculated with the same inactivated SVCV vaccine in the same dosage in Env38 knockout (Env38$^{-/-}$) zebrafish. 35 days later, all the immunized groups and unimmunized control groups were challenged with living SVCV ($10^5$ TCID$_{50}$). In the next 10 days, the mortality of each group was recorded, and the statistics of survival were analyzed. The viral load in the WT and knockout zebrafish was measured by viral titer (TCID$_{50}$) in EPC cells. Viral samples were prepared from the homogenates of whole zebrafish. For this procedure, the homogenates were centrifuged (3,000 g) at 4˚C for 30 min, then the supernatants were collected and filtered through 0.22 µm membrane (Millipore Express). The viral titer (TCID$_{50}$) was examined by using the Reed-Muench method as described above [77].

## Functional evaluation for IFNφ1 in inducing *env38* expression

Zebrafish were i.p. stimulated with SVCV ($10^5$ TCID$_{50}$) to induce the production of type I IFNφs. Meantime, the fish was administered with mouse anti-IFNφ1 Ab (10 µg/fish) three times at 24 h intervals for IFNφ1 neutralizing. In parallel, a rescue assay was performed by i.p.

administering recombinant zebrafish IFNφ1 protein (10 μg/fish) back into the IFNφ1-neutralized fish. Additionally, zebrafish in the control groups were treated with isotype mouse IgG and rescued with mock PBS. The effect of IFNφ1 on Env38 expression was determined by FCM analysis (FACSCalibur) and RT-qPCR at protein and mRNA levels, based on the percentage of Env38$^+$ cells and transcriptional level of *env38*, *isg15* and *viperin* mRNAs in leukocytes from spleen, head kidney and peripheral blood as described above.

## Statistical analysis

Statistical analysis and graphical presentation were performed with GraphPad Prism and Origin Pro. The results were expressed as mean ± standard error of the mean (SEM). Statistical evaluation of differences was assessed using one-way ANOVA, followed by an unpaired two-tailed t-test. Survival curve differences were assessed using the log-rank test. Statistical significance was defined as $^*p<0.05$, $^{**}p<0.01$, $^{***}p<0.001$, ns, no significant difference. All experiments were replicated at least three times. The data that support the findings of this study are deposited in the Dryad repository: https://doi.org/10.5061/dryad.t4b8gtj5c [78].

## Supporting information

**S1 Data. Excel spreadsheet containing, in separate sheets, the underlying numerical data and statistical analysis for Figs 5A, 6A, 6B, 7B, 7D, 8B, 8E, 8F, 9A, 9B, 9E, 10K, 12C, 12E, S3A, S3D, S3G and S7.**
(XLSX)

**S1 Fig. The cDNA and amino acid sequences of Env38 and multiple alignment of Env38 with other retroviral envelope proteins.** (A) The cDNA sequence of *env38* gene and the deduced amino acids of Env38 protein. The non-coding regions represent the LTR, in which the U3, R and U5 sequences were underlined with the red line, purple line and green line, respectively. The signal peptide and transmembrane domain of the Env38 protein were bottomed with red and purple, respectively. The sequences forming conserved disulfide bonds were bottomed with green. The potential SU-TM cleavage site was bottomed with blue. Seven aspartic acids forming deduced N-glycosylation sites in Env38 protein were framed with red. The asterisk represented the stop codon. (B) Multiple alignment of the amino acid sequence of Env38 with a previously described ZFERV Env protein of zebrafish and Syncytin proteins of mammals. The relatively conserved motifs or amino acids in the protease cleavage site (CS), the fusion peptide (FP), the N-terminal heptad repeats (NHR), the C-terminal heptad repeats (CHR) and the linker motif between NHR and CHR (CX6CC) were marked with different colors. The Genbank accession numbers of the sequences were as follows: Homo sapiens-Syncytin-1, NP_001124397.1; Homo sapiens-Syncytin-2, NP_997465.1; Mus musculus-Syncytin-A, NP_001013773.1; ZFERV Env, AAM34209.1.
(TIF)

**S2 Fig. The cDNA and deduced amino acid sequences of zebrafish *furina* and *furinb* genes.** The signal peptides and transmembrane domains were bottomed with red and purple, respectively. The FU domains with furin-like cysteine rich regions were bottomed with blue.
(TIF)

**S3 Fig. Examination on the knockdown efficacy of siRNA-delivery lentivirus.** (A) RT-qPCR analysis for the efficacy of siRNAs targeting *env38* mRNA in HEK293T cells. pSUPER construct containing the scrambled siRNA was used as the negative control. (B and C) Examination on the titers of the constructed Env38siRNA-LV in HEK293T cells based on EGFP fluorescence observed under a fluorescence microscope (B) and analyzed by FCM analysis (C).

Fluorescence images were captured using Zeiss Axiovert 40 CFL with $100 \times$ original magnification. (D-F) Examination on the efficiency of Env38siRNA-LV-mediated knockdown of Env38 in leukocytes from spleen, head kidney and peripheral blood via RT-qPCR (D), FCM analysis (E) and in spleen tissue via Western blot (F) under SVCV ($10^5$ TCID$_{50}$) stimulation. (G) Grayscale quantization of Env38 protein examined by Western blot, in which the normalization was complied by the gray value ratio of the target Env38 protein and the internal reference GAPDH protein. Nonrelated control groups were administered with scrambled siRNA-LV. Negative control groups received mock PBS. RT-qPCRs were run in combination with the endogenous β-actin control. Error bars represented SEM. ($^*p < 0.05$; $^{**}p < 0.01$; $^{***}p < 0.001$; ns, no significant difference).
(TIF)

**S4 Fig. Identification of Env38 knockout zebrafish (Env38$^{-/-}$ zebrafish).** (A) Sequencing chromatograms of *env38* nucleicacid sequence in knockout zebrafish. The shadows presented the alternative bases between wild type (WT) and knockout (KO) zebrafish. The targeted "C" was replaced by "TTCAAGGCT". (B) Western blot analysis of Env38 protein in spleen tissues of WT and KO zebrafish.
(TIF)

**S5 Fig. Subcellular localization analysis of Env38 by anti-Env38 antibody.** HEK293T cells were transfected with the recombinant expression plasmid of pcDNA3.1-Flag-Env38-LTR (0.6 μg/ml) for 48 h, and then fixed and labeled with mouse anti-Env38 Ab (1:500), followed by FITC-conjugated goat anti-mouse IgG (1:1,000). Next, the cells were stained with the cell membrane probe DiI (A), ER-tracker (B) or Golgi-tracker (C) and nuclei probe DAPI. The blue, green, and red fluorescence images showed DAPI-labeled nuclei, FITC-labeled Env38 protein, and DiI-labeled cell membrane, ER-tracker-labeled ER, and Golgi-tracker-labeled Golgi apparatus. Fluorescence images were captured using a Laser scanning confocal microscope (FV3000) with $60 \times$ oil glass.
(TIF)

**S6 Fig. Immunofluorescence staining for the distribution of Env38.** (A and B) Immunofluorescence staining for the distribution of Env38 on MHC-II$^+$ and IgM$^+$ cells in leukocytes from spleen, head kidney and peripheral blood of zebrafish with SVCV ($10^5$ TCID$_{50}$) stimulation. Cells were stained with mouse anti-Env38 Ab (1:500) together with rabbit anti-MHC-IIα Ab (1:500) (A) or mouse anti-Env38 Ab (1:500) together with rabbit anti-IgM Ab (1:500) (B), followed by Alexa Fluor 594-conjugated goat anti-mouse IgG (1:1,000) and FITC-conjugated goat anti-rabbit IgG (1:1,000), respectively. (C and D) Immunofluorescence staining for the absence of Env38 protein in leukocytes of zebrafish without SVCV stimulation. The leukocytes were sorted from spleen, head kidney and peripheral blood and fixed and then permeabilized with Saponin (C) or not (D). Next, the leukocytes were labled with mouse anti-Env38 Ab (1:500) and rabbit anti-MHC-IIα Ab (1:500), followed by Alexa Fluor 594-conjugated goat anti-mouse IgG (1:1,000) and FITC-conjugated goat anti-rabbit IgG (1:1,000). DAPI stain showed the location of the nuclei. Fluorescence images were captured using a Laser scanning confocal microscope (FV3000) with $60 \times$ oil glass.
(TIF)

**S7 Fig. Transcriptional expression analysis of the *env38* gene in zebrafish tissues and leukocytes.** The kinetic expression patterns of the *env38* in spleen, head kidney tissues and leukocytes from zebrafish under stimulation with SVCV ($10^5$ TCID$_{50}$) were examined by RT-qPCR. Each sample was obtained from at least 10 fish. RT-qPCRs were run in combination with the endogenous β-actin control. Error bars represented SEM. ($^*p < 0.05$; $^{**}p < 0.01$; $^{***}p < 0.001$;

ns, no significant difference).
(TIF)

**S8 Fig. Schematic diagram of zebrafish CD4 and MHC-IIα proteins and their truncated forms.** (A) Schematic diagram of the wild type CD4, truncated proteins with deletions of IgG domains of CD4 and the truncated extracellular CD4-D2 domain protein (sCD4-D2). (B) Schematic diagram of the wild type and truncated structures of MHC-IIα proteins. The architecture analysis of CD4 and MHCIIα proteins were conducted by SMART program. (C) Western blot analysis of the recombinant sCD4-CD2 protein with anti-Flag Ab (1:5,000) from supernatant of HEK293T cells transfected with the pcDNA3.1-His-Flag-CD4-D2 recombinant constructs or an empty control construct.
(TIF)

**S1 Table. Primers used in this study.**
(XLSX)

## Acknowledgments

We greatly thank Prof. Yibing Zhang of Institute of Hydrobiology, Chinese Academy of Sciences for providing the SVCV strain and EPC cell line.

## Author Contributions

**Conceptualization:** Li-xin Xiang, Jian-zhong Shao.

**Data curation:** Yun Hong, Chong-bin Hu, Jian-zhong Shao.

**Formal analysis:** Yun Hong, Chong-bin Hu, Jun Bai.

**Funding acquisition:** Yun Hong, Li-xin Xiang, Jian-zhong Shao.

**Investigation:** Yun Hong, Chong-bin Hu, Jun Bai, Dong-dong Fan, Ai-fu Lin, Jian-zhong Shao.

**Methodology:** Yun Hong, Chong-bin Hu, Li-xin Xiang, Jian-zhong Shao.

**Project administration:** Yun Hong, Li-xin Xiang, Jian-zhong Shao.

**Resources:** Dong-dong Fan, Ai-fu Lin, Li-xin Xiang, Jian-zhong Shao.

**Supervision:** Li-xin Xiang, Jian-zhong Shao.

**Validation:** Yun Hong, Jian-zhong Shao.

**Visualization:** Yun Hong.

**Writing – original draft:** Yun Hong, Jian-zhong Shao.

**Writing – review & editing:** Yun Hong, Chong-bin Hu, Jian-zhong Shao.

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
