## [Decision Letter · Decision Letter 0]

25 Jul 2022

Dear Professors Shao and Xiang,

Thank you very much for submitting your manuscript "Essential role of an ERV-derived Env38 protein in adaptive humoral immunity against an exogenous SVCV infection in a zebrafish model" for consideration at PLOS Pathogens. As with all papers reviewed by the journal, your manuscript was reviewed by members of the editorial board and by several independent reviewers. In light of the reviews (below this email), we would like to invite the resubmission of a significantly-revised version that takes into account the reviewers' comments.

I am returning your manuscript with three reviews, all of which point out the overall quality and the originality of the findings reported in your manuscript. The reviewers came to different conclusions about the paper, as you will see. After reading the reviews and looking at the manuscript, I recommend Major Revision based on the critiques from the more critical reviews. I am sorry I cannot be more positive at the moment, however we are looking forward to receiving your revision. With additional work, the manuscript will be suitable for a resubmission, if you so wish to do so. Note that we may send your paper back to some of the more critical reviewers upon resubmission.

Please pay particular attention to the following reviewer suggestions and give them due consideration:

Possible artifacts about the subcellular localization of ENV38, due to the use of fusion proteins, is a major issue. If possible, please provide date from untagged protein detected using the ENV38-specific antibodies that you generated. If not, the use of C-terminally tagged proteins, as suggested by reviewer 1, is a suitable alternative.

In this regard, as an editorial request, figure panels should always explicitly indicate detection of tags; e.g., the second panel of fig3A should have been labelled “GFP-ENV38” instead of “ENV38”.

A control for figure 4C/D, with cells from control animals, should be provided.

The survival curve of unimmunized KO animals should be included on fig 8A.

The issue of western blot lanes that may have been stitched from different blots is also important. Please provide image of the entire blots in your rebuttal letter; this would not be needed in the final manuscript.

We cannot make any decision about publication until we have seen the revised manuscript and your response to the reviewers' comments. Your revised manuscript is also likely to be sent to reviewers for further evaluation.

Sincerely,

Jean-Pierre Levraud, PhD

Guest Editor

PLOS Pathogens

Adolfo García-Sastre

Section Editor

PLOS Pathogens

Kasturi Haldar

Editor-in-Chief

PLOS Pathogens

orcid.org/0000-0001-5065-158X

Michael Malim

Editor-in-Chief

PLOS Pathogens

orcid.org/0000-0002-7699-2064

Reviewer's Responses to Questions

**Part I - Summary**

Reviewer #1: The manuscript demonstrated the function of Env38 in zebrafish adaptive immunity against SVCV infection. Env38 is a glycosylated membrane protein mainly distributed on MHC-II+ APCs, which could be induced by zebrafish IFNφ1. Through a series of molecular and biochemical analyses from in vivo and in vitro experiments, the authors found that Env38 interacted with both MHC-IIα of APCs and CD4 of CD4+ T cells to form MHC-TCR-CD4 complexes, which activated CD4+ T cells, and promoted the proliferation of IgM+/IgZ+ B cells. The authors are the first to discover the role of Env protein in adaptive humoral immunity, and the results are well presented.

Reviewer #2: Accumulating evidence suggests that endogenous retroviral elements have been co-opted by host cells. Here, Hong and colleagues describe a fascinating example of how a retroviral envelope protein (termed Env38) mediates adaptive immune responses against Spring viraemia of carp virus (SVCV) in zebra fish. The authors convincingly show that Env38 expression is induced in response to SVCV infection and IFNφ1 stimulation. Using an elegant combination of blocking, knock-down and knockout experiments, the authors show that Env38 is required for efficient SVCV-induced proliferation of CD4 + T cells and IgM + /IgZ + B cells, as well as IgM/IgZ antibody production. Notably, the authors also demonstrate that Env38 cross-links MHC-II and CD4, and they identify the protein domains involved in this interaction. One major strength of the study is the use of several alternative approaches (e.g. antibody-mediated blocking, knockdown and knockout experiments) and the validation of in vitro findings in zebra fish models in vivo. The manuscript contains an impressive amount of data. While I am listing quite a number of points and detailed comments below, most of my concerns can be addressed by textual changes.

Reviewer #3: In this study, the authors investigate the role of an envelope protein (Env38) derived from an endogenous retrovirus (ERV) in zebrafish adaptive immunity against Spring viraemia

of carp virus (SVCV). They observe that Env38 is a glycosylated membrane protein mainly distributed on MHC-II+ antigen-presenting cells. Antibody blockade and Env38 knockdown and knockout assays show that disruption of Env38 expression or function impairs the activation of SVCV-induced CD4+ T cells and zebrafish defense against SVCV challenge. The authors present data showing that Env38 activates CD4+ T cells by promoting the formation of a pMHC-TCR-CD4 complex via cross-linking MHC-II and CD4 molecules between the APCs and CD4+ T cells. The expression of Env38 appears to be strongly induced by zebrafish interferons. The study implicates Env38 in host immune defense against an exogenous virus.

Overall, the study represents a large amount of high-quality work that provides new insights into the role of endogenous retroelements in adaptive immunity. In my view, the study will be of interest to a wide audience. I have mostly minor comments for improvement of the manuscript.

**Part II – Major Issues: Key Experiments Required for Acceptance**

Reviewer #1: The manuscript is very informative, which is a collection of too vast an array of different findings. Please re-write it with a clear story. Such as these results involved in the glycosylation and cleavage-processing of Env38 distracted my attention. In fact, the authors did not provide these data that the glycosylation and cleavage-processing of Env38 was involved in the function of Env38 in host adaptive immunity.

Reviewer #2: 1. Fig. 3: For the localization analyses shown in this figure, Env38 was cloned into pEGFPC1, resulting in an EGFP-Env38 fusion protein. It is highly likely that fusion of EGFP to the N-terminus of Env38 affects the normal subcellular localization of the latter. Furthermore, the N-terminal EGFP may also be cleaved off if it was indeed fused to the N-terminal signal peptide of Env. The authors should use untagged or C-terminally tagged (His-tag, FLAG-tag, etc.) Env38 instead of EGFP-Env38 chimeras to determine the subcellular localization of Env38 in the presence and absence of SVCV.

Fig. 4C/D: Can the authors include a control without SVCV stimulation to validate the relocalization of Env38 they propose? A comparison of permeabilized and unpermeabilized cells may also be helpful in this case.

2. Lines 274/275: The authors state that “Env38 might act as a trafficking transmembrane protein that underwent post translational glycosylation […] after SVCV stimulation”. Western blotting needs to be performed to validate the hypothesis that SVCV stimulation alters the glycosylation of Env38.

3. Fig. 8A: Can the authors include a survival curve of unimmunized KO animals?

Reviewer #3: (No Response)

**Part III – Minor Issues: Editorial and Data Presentation Modifications**

Reviewer #1: 1) In Fig 2E, Furin enzymes could cleave Env38 into SU and TM subunits, why WB can only detect TM subunit but not SU subunit? In Fig 2F, how to explain the undetectable expression of furina and furinb in Env38+ cells?

2) Lines 301 and 1162: How did the authors conclude that the leukocytes they isolated were lymphatic and myeloid leukocytes?

3) In Fig 5B, the IgM produced in SVCV+contsiRNA group was higher than that of SVCV group, and the mRNA level of Env38 in SVCV group was higher than SVCV+contsiRNA group. How to interpret these results?

4) In Fig 8A: To confirm that these zebrafish died from SVCV infection, the authors should also investigate the viral loads in the wildtype and knockout zebrafish.

5) In Fig 9G, the Env38 protein band expressed in E.coli (48 kDa) should be smaller than that expressed in sf9 cells (60 kDa).

6) Line 335: Please define CD4+ Tsvcv when it first appears.

Reviewer #2: 1. Lines 257-259: The authors state that „minimal cleavage of the Env38 protein was detected in the spleen, head kidney, and leucocytes of zebrafish stimulated with SVCV (105 TCID50) (Fig 2B).” However, no such band is visible in Fig. 2B. Can the authors show the same blot in a higher intensity to make the cleaved 25 kDa form visible?

2. Fig. 2: In many cases, individual lanes were stitched together and it remains unclear whether they originate from the same blot. In Fig. 2C, for example, it remains unclear whether untreated and tunicamycin treated HEK293T cell samples were run on the same blot. Furthermore, the PNGase F treated Sf9 lysates should be shown next to an untreated Sf9 negative control. Instead, they are shown next to an untreated HEK293T cell sample, which may be misleading. Similarly, it remains unclear whether the samples in Fig. 2E were run on the same blot. In this case, it would have been nice to validate expression of Furin in lanes 2 and 3 instead of separate samples. Finally, detection of a housekeeping protein (e.g. GAPDH or beta-actin) may enable a better comparison of the samples in Figs. 2C and 2E.

3. Line 46: It has been proposed that endogenous retroviruses may also be found in plants (e.g., Laten and Gaston, Plant Transposable Elements, Topics in Current Genetics 24, DOI 10.1007/978-3-642-31842-9_6, 2012). Thus, statements regarding the exclusive presence of ERVs in vertebrates may be toned down accordingly.

4. Lines 206-209: The description of zebrafish IFNφ and sCD4ΔD2 may be mentioned at a later time point, when describing the experiments shown in Fig. 10 and S7.

5. Line 207: The authors are referring to “CD4-D2”, rather than “CD4ΔD2”, I assume.

6. Fig. S2B: A negative control may be included (optional)

7. Line 242: “inhibition of glycosylation” may be more appropriate than “deglycosylation” in the case of tunicamycin.

8. Line 260, Fig. 2F: What kind of Env38+ cells are shown here?

9. Line 265: The authors may briefly explain what EPC cells are and why they were used here.

10. The panels in Fig. 4 should be re-ordered according to their appearance and discussion in the text. In general, figures should be numbered according to the text.

11. Fig. 4E, 6C, 6E, 7A, 7C, 7D, 8B, 8C, 10A, 10B, 10D: Isotype controls should be included (optional).

12. Fig. 9I, J: Detection of FLAG-Env in the IP samples is missing. This control is required to demonstrate similar pull-down efficiencies.

13. Lines 342/343: “the percentage of CD4+ T cells in Env38-blockade groups was comparable to that of the negative control groups”. Better rephrase. The percentage in the Env38-blockade group was two-fold higher than that in the negative control.

Reviewer #3: 1. Line 209: Figs. 10F and S7B are referred to out of order.

2. Fig. 2. It is surprising that zebrafish but not human furin is active in cleaving Env38, given that this protein contains what appears to be a consensus furin cleavage site. Perhaps the authors could comment.

3. Line 327, 370 and elsewhere: the authors state that Env38 plays an “essential” role in zebrafish antiviral innate immunity. “Essential” is an overstatement.

4. Fig. 9 is confusing. I think the basis for the confusion is that the left sides of the panels are mis-labeled “IP” (immunoprecipitation) instead of “IB” (immunoblot) as described in the methods. This should be corrected/clarified, and the figure legend should clearly explain what was done and how the figure is labeled.

5. Throughout the paper, the English is in need of editing. Most errors are simple grammatical mistakes, but in other places the text is difficult to understand. For example, line 267 states that cells were “signed” when I think the authors mean “stained”; line 577, what is meant by 6HB “texture”? Line 577, the sentence “the fusion peptide was locally hydrophobic, implying that Env38 was fusion defective even though the 6HB post-fusion structure was still retained” doesn’t make sense. Maybe the authors mean “locally hydrophilic”? Line 590 “non-cleavage reason”?

6. In the Discussion section, the authors speculate on recognition of the furin cleavage site. If Env38 cleavage is disadvantageous to the host, why did Env38 not simply lose the furin cleavage site?

PLOS authors have the option to publish the peer review history of their article (what does this mean?). If published, this will include your full peer review and any attached files.

Reviewer #1: No

Reviewer #2: No

Reviewer #3: No
---

## [Decision Letter · Decision Letter 1]

5 Oct 2022

Dear Professors Shao and Xiang,

Thank you very much for resubmitting your revised version of manuscript “Essential role of an ERV-derived Env38 protein in adaptive humoral immunity against an exogenous SVCV infection in a zebrafish model” (PPATHOGENS-D-22-00798_R1) for review by PLoS Pathogens. Your manuscript has been re-evaluated by the three reviewers that had assessed the original ones, and all were very positive. One of them, however, has pointed out an aspect of the manuscript that should be improved. We therefore ask you to modify the manuscript according to the review recommendations before we can consider your manuscript for acceptance.

I am returning your manuscript with three reviews. After reading the reviews and looking at the manuscript, I have decided that the minor revision requested by reviewer 2, regarding interpretation of Figure 9A, needs to be done to prepare the manuscript for publication. Please also answer his requests for clarification regarding the source images of the western blots displayed on Figure 2. Please give these issues due consideration; if they are addressed, I hope to be able to make a final decision without sending the manuscript out for a second round of review.

Sincerely,

Jean-Pierre Levraud, PhD

Guest Editor

PLOS Pathogens

Adolfo García-Sastre

Section Editor

PLOS Pathogens

Kasturi Haldar

Editor-in-Chief

PLOS Pathogens

orcid.org/0000-0001-5065-158X

Michael Malim

Editor-in-Chief

PLOS Pathogens

orcid.org/0000-0002-7699-2064

Dear Professors Shao and Xiang,

Thank you very much for resubmitting your revised version of manuscript “Essential role of an ERV-derived Env38 protein in adaptive humoral immunity against an exogenous SVCV infection in a zebrafish model” (PPATHOGENS-D-22-00798_R1) for review by PLoS Pathogens. Your manuscript has been re-evaluated by the three reviewers that had assessed the original ones, and all were very positive. One of them, however, has pointed out an aspect of the manuscript that should be improved. We therefore ask you to modify the manuscript according to the review recommendations before we can consider your manuscript for acceptance.

I am returning your manuscript with three reviews. After reading the reviews and looking at the manuscript, I have decided that the minor revision requested by reviewer 2, regarding interpretation of Figure 9A, needs to be done to prepare the manuscript for publication. Please also answer his requests for clarification regarding the source images of the western blots displayed on Figure 2. Please give these issues due consideration; if they are addressed, I hope to be able to make a final decision without sending the manuscript out for a second round of review.

Sincerely,

Jean-Pierre Levraud, PhD

Guest Editor

PLOS Pathogens

Adolfo García-Sastre

Section Editor

PLOS Pathogens

Kasturi Haldar

Editor-in-Chief

PLOS Pathogens orcid.org/0000-0001-5065-158X

Grant McFadden

Editor-in-Chief

PLOS Pathogens orcid.org/0000-0002-2556-3526

Reviewer Comments (if any, and for reference):

Reviewer's Responses to Questions

**Part I - Summary**

Reviewer #1: (No Response)

Reviewer #2: In the revised version of their manuscript, Hong and colleagues have addressed all of my initial comments. In addition to several textual changes, the authors have also included additional experimental data:

- Since an N-terminal EGFP tag may affect the subcellular localization of Env38 (or simply be cleaved off), the authors also monitored the localization of C-terminally tagged and untagged Env38. These new data confirm the authors’ initial conclusions, and the respective images have been included in the revised version of their manuscript.

- Importantly, Hong and colleagues have also included a survival curve of unimmunized KO animals in Fig. 9A. This important control demonstrates that deletion of Env38 also significantly reduces survival in non-immunized animals. Thus, Env38 KO does not specifically attenuate protection upon immunization, and the statements in lines 397-403 need to be changed accordingly.

- The authors have also detected FLAG-Env in the IP samples in Fig. 10I to demonstrate similar pull-down efficiencies.

- Finally, the authors have re-run several of their Western blot and loaded all samples together with their corresponding controls. The original images of the Env38 cleavage blot (page 22 of the rebuttal) suggest that Env38 and GAPDH were not detected on the same membrane since 3 and 5 lanes are shown, respectively. This should be clarified. Furthermore, it still remains unclear whether Furin expression (Fig. 2E) and Env38 cleavage (Fig. 2D) were detected in the same samples/blots.

Overall, however, the manuscript has significantly improved and the authors’ main conclusion are supported by the data shown.

Reviewer #3: N/A

**Part II – Major Issues: Key Experiments Required for Acceptance**

Reviewer #1: (No Response)

Reviewer #2: None

Reviewer #3: N/A

**Part III – Minor Issues: Editorial and Data Presentation Modifications**

Reviewer #1: (No Response)

Reviewer #2: As mentioned above, I feel that the authors need to change their interpretation of Fig. 9A, given the differences in survival between WT and KO animals in the absence of immunization.

Reviewer #3: the authors have addressed my comments.

PLOS authors have the option to publish the peer review history of their article (what does this mean?). If published, this will include your full peer review and any attached files.

Reviewer #1: **Yes: **Yes

Reviewer #2: No

Reviewer #3: No

Figure Files:

Data Requirements:

Reproducibility:

References:

---

## [Editor Report · Decision Letter 2]

5 Jan 2023

Dear Professors Shao and Xiang,

Thank you for the additional material provided in support of your revised article “Essential role of an ERV-derived Env38 protein in adaptive humoral immunity against an exogenous SVCV infection in a zebrafish model” (PPATHOGENS-D-22-00798_R2) for review to PLoS Pathogens.

Unfortunately, because the edges of the blots are not detectable on the scanned images that you have provided in support of Figure 2, we have not been able to reach a definitive conclusion regarding data integrity.

In these conditions, we cannot accept the article in the present form but we decided to allow you to repeat these experiments.

We thus request that you generate a new version of figure 2, from a replicate experiment, and provide the underlying blots to allow for proper inspection.

We cannot make any decision about publication until we have seen the revised manuscript and your response to these requests. Your revised manuscript is also likely to be sent to reviewers for further evaluation.

Sincerely,

Jean-Pierre Levraud, PhD

Guest Editor

PLOS Pathogens

Adolfo García-Sastre

Section Editor

PLOS Pathogens

Kasturi Haldar

Editor-in-Chief

PLOS Pathogens

orcid.org/0000-0001-5065-158X

Michael Malim

Editor-in-Chief

PLOS Pathogens

orcid.org/0000-0002-7699-2064

Dear Professors Shao and Xiang,

thank you for the additional material provided in support of your revised article “Essential role of an ERV-derived Env38 protein in adaptive humoral immunity against an exogenous SVCV infection in a zebrafish model” (PPATHOGENS-D-22-00798_R2) for reveiw to PLoS Pathogens.

Unfortunately, because the edges of the blots are not detectable on the scanned images that you have provided in support of Figure 2, we have not been able to reach a definitive conclusion regarding data integrity.

In these conditions, we cannot accept the article in the present form but we decided to allow you to repeat these experiments.

We thus request that you generate a new version of figure 2, from a replicate experiment, and provide the underlying blots to allow for proper inspection.
---

## [Editor Report · Decision Letter 3]

20 Feb 2023

Dear Prof. Shao,

We are pleased to inform you that your manuscript 'Essential role of an ERV-derived Env38 protein in adaptive humoral immunity against an exogenous SVCV infection in a zebrafish model' has been provisionally accepted for publication in PLOS Pathogens.

Best regards,

Jean-Pierre Levraud, PhD

Guest Editor

PLOS Pathogens

Adolfo García-Sastre

Section Editor

PLOS Pathogens

Kasturi Haldar

Editor-in-Chief

PLOS Pathogens

orcid.org/0000-0001-5065-158X

Michael Malim

Editor-in-Chief

PLOS Pathogens

orcid.org/0000-0002-7699-2064

The new version of the figure 2, together with the underlying blots, is now satisfactory. Thank you.
---

## [Editor Report · Acceptance letter]

14 Mar 2023

Dear Prof. Shao,

We are delighted to inform you that your manuscript, "Essential role of an ERV-derived Env38 protein in adaptive humoral immunity against an exogenous SVCV infection in a zebrafish model," has been formally accepted for publication in PLOS Pathogens.

Best regards,

Kasturi Haldar

Editor-in-Chief

PLOS Pathogens

orcid.org/0000-0001-5065-158X

Michael Malim

Editor-in-Chief

PLOS Pathogens

orcid.org/0000-0002-7699-2064